# Aging-related upregulation of the homeobox gene *caudal* represses intestinal stem cell differentiation in *Drosophila*

Kun Wu[1ʘ], Yiming Tang[1ʘ], Qiaoqiao Zhang[1ʘ], Zhangpeng Zhuo[1], Xiao Sheng[2], Jingping Huang[1], Jie'er Ye[1], Xiaorong Li[1], Zhiming Liu[2], Haiyang Chen[iD][2]*

**1** Key Laboratory of Gene Engineering of the Ministry of Education, State Key Laboratory of Biocontrol, School of Life Sciences, Sun Yat-sen University, Guangzhou, Guangdong, China, **2** Laboratory for Aging and Stem Cell Research, National Clinical Research Center for Geriatrics, West China Hospital, Sichuan University, Chengdu, Sichuan, China

ʘ These authors contributed equally to this work.
* chenhy82@scu.edu.cn

**Data Availability Statement:** The RNA-seq data that support the findings of this study have been deposited in the Sequence Read Archive (SRA) under BioProject accession PRJNA686687. Source

## Abstract

The differentiation efficiency of adult stem cells undergoes a significant decline in aged animals, which is closely related to the decline in organ function and age-associated diseases. However, the underlying mechanisms that ultimately lead to this observed decline of the differentiation efficiency of stem cells remain largely unclear. This study investigated *Drosophila* midguts and identified an obvious upregulation of *caudal* (*cad*), which encodes a homeobox transcription factor. This factor is traditionally known as a central regulator of embryonic anterior-posterior body axis patterning. This study reports that depletion of *cad* in intestinal stem/progenitor cells promotes quiescent intestinal stem cells (ISCs) to become activate and produce enterocytes in the midgut under normal gut homeostasis conditions. However, overexpression of *cad* results in the failure of ISC differentiation and intestinal epithelial regeneration after injury. Moreover, this study suggests that *cad* prevents intestinal stem/progenitor cell differentiation by modulating the Janus kinase/signal transducers and activators of the transcription pathway and Sox21a-GATAe signaling cascade. Importantly, the reduction of *cad* expression in intestinal stem/progenitor cells restrained age-associated gut hyperplasia in *Drosophila*. This study identified a function of the homeobox gene *cad* in the modulation of adult stem cell differentiation and suggested a potential gene target for the treatment of age-related diseases induced by age-related stem cell dysfunction.

## Author summary

Adult stem cells undergo an aging-related decline of differentiation efficiency in aged animals. However, the underlying mechanisms that ultimately lead to this observed decline of differentiation efficiency in stem cells still remain largely unclear. By using the *Drosophila* midgut as a model system, this study identified the homeobox family transcription factor gene *caudal* (*cad*), the expression of which is significantly upregulated in intestinal stem cells (ISCs) and progenitor cells of aged *Drosophila*. Depletion of *cad* promoted

data of Fig 4A have been provided as S1 Table and S2 Table.

**Funding:** This work was supported by the National Key R&D Program of China (2020YFA0803602 and 2018YFA0108301), the National Natural Science Foundation of China (31622031, 31671254, and 91749110) (H.C.), the Guangdong Natural Science Funds for Distinguished Young Scholars (2016A030306037) (H.C.), the National Clinical Research Center for Geriatrics, West China Hospital, Sichuan University (Z2020201006) (H.C.), and the 1.3.5 project for disciplines of excellence, West China Hospital, Sichuan University (H.C.). The funders had no role in study design, data collection and analysis, decision to publish, or preparation of the manuscript.

**Competing interests:** The authors have declared that no competing interests exist.

quiescent ISCs to become activate and produce enterocytes (ECs) in midguts under normal gut homeostasis conditions; However, overexpression of *cad* resulted in the failure of ISC differentiation and intestinal epithelial regeneration after injury. Moreover, *cad* prevents ISC-to-EC differentiation by inhibiting JAK/STAT signaling, and the expressions of *Sox21a* and *GATAe*. Reduction of *cad* expression in intestinal stem/progenitor cells restrained age-associated gut hyperplasia in *Drosophila*. These findings enable a detailed understanding of the roles of homeobox genes in the modulation of adult stem cell aging in humans. This will be beneficial for the treatment of age-associated diseases that are caused by a functional decline of stem cells.

## Introduction

In metazoans, adult stem cells are responsible for the maintenance of long-term homeostasis of many organs or tissues through their ability to produce terminal differentiated cells to replace lost or damaged cells. Under homeostatic conditions, adult stem cells are largely quiescent, but will promptly be activated to proliferation and differentiation upon tissue injury [1]. However, when animals get old, both the self-renewal and differentiation efficiency of these resident stem cells exhibit significant declines [2]. Many studies have demonstrated that the functional decline of organs can be clearly delayed or even reversed when their resident stem cells are not subject to aging-related functional decline [3]. Preventing the functional decline of adult stem cells is therefore a promising area of anti-aging research. However, because of the complexities of the stem cell niche and external environment, the underlying molecular mechanisms modulating the aging-related functional decline of adult stem cells remain largely unknown.

The midgut of adult *Drosophila* has emerged as a powerful model system for the study of mechanisms underlying the age-related decline in stem cell function [4–6]. Because of its simple cellular components and wide array of available genetic tools for the *Drosophila* midgut, this system has been used to identify potential strategies to reverse or delay the dysfunction of stem cells in aged animals [7–9]. Similar to the mammalian intestine, the epithelium of the *Drosophila* midgut is maintained by resident intestinal stem cells (ISCs) with the ability to self-renew and differentiate into terminal function cells [5,6,10–12]. *Drosophila* ISCs reside in the basement membrane of the midgut epithelium and specifically express the transcription factor Escargot (Esg) and the Notch ligand Delta (Dl). The ISCs in the *Drosophila* midgut undergo asymmetric cell division to produce the new ISCs and progenitor cells. When asymmetric dividing of ISCs occurs, ISCs can produce either enteroendocrine mother cells (EMCs, which further differentiate into secretory enteroendocrine cells (EEs)) or non-dividing enteroblasts (EBs, which further differentiate into absorptive enterocytes (ECs)) depending on Notch activity (S1A Fig) [5,6,13–16]. A high level of Notch signaling drives ISCs to produce ECs, while a low level of Notch signaling drives ISCs to produce EEs [14,15]. Under normal homeostatic conditions, most ISCs are in a quiet state and proliferate slowly in young *Drosophila*. Thus, the numbers of ISCs and progenitor cells are relatively low and remain stable in the midguts of young and healthy *Drosophila*. However, after injury, these ISCs are rapidly activated in response to damage-induced signals, and temporarily increase their proliferation rates, thus initiating EC and/or EE production [12,17,18]. In aged *Drosophila*, many ISCs in the midgut are abnormally activated [19–21], and the differentiation capacity of ISCs and EBs follows a continuous decrease with aging [9,20]. This results in the accumulation of Delta positive ($Dl^+$) expressing cells and Esg positive ($Esg^+$) cells in the midgut of aged *Drosophila* [19,20,22].

Studies have shown that multiple signaling pathways, including reactive oxygen species (ROS) signaling [23,24], endoplasmic reticulum stress-induced signaling [25,26], c-Jun N-terminal kinase (JNK) signaling [20], insulin signaling [27–29], p38-MAPK signaling [22], DGF/VEGF signaling [19], and mTOR signaling [30], play roles in the modulation of aging-related ISC hyper-proliferation. Moreover, preventing the accumulation of $Esg^+$ cells in the midgut obviously extended the lifespan of *Drosophila* [3]. However, the regulators and signaling pathways that regulate the aging-related decline of the differentiation capacity of ISCs and EBs remain largely unclear.

Homeobox family transcription factors, which exhibit DNA-binding transcription factor activity, are the central regulators of embryonic anterior-posterior body axis pattern formation during development [31]. *Caudal* (*cad*) homologs, which belong to the homeobox transcription factor family, have been identified in organisms ranging from *Caenorhabditis elegans* to humans [32–34]. These homologs have been found to play important roles in intestinal development and maintenance [35]. In the mouse gastrointestinal tract, the mammalian homologs of Cad, Caudal type homeobox 1 (Cdx1) and Caudal type homeobox 2 (Cdx2), have been reported to perform critical roles in the regulation of the developmental process of the intestine [36,37]. Moreover, dysregulations of Cdx1 and Cdx2 have been shown to be involved in the carcinogenesis of gastrointestinal cancers [38]. In addition to its function in the establishment of the anteroposterior axis during early embryogenesis, the *Drosophila* homeobox gene *cad* has been shown to regulate gut development [39,40]. It has been demonstrated that *cad* has very important functions in the innate immune response in the *Drosophila* midgut [41,42]. Recently, *cad* has been found to be upregulated in the posterior midgut of aged *Drosophila*, where it regulates gut immune homeostasis [43]. However, this study [43] did not present the cell-type specific expression pattern of *cad*. The function of cad in intestinal stem/progenitor cells and gut epithelial homeostasis remains unknown.

Using the *Drosophila* midgut as a model system, this study found that the expression of *cad* increases in ISCs and EBs of aged *Drosophila*. Furthermore, high levels of *cad* expression in ISCs and EBs prevent the ISC-to-EC differentiation through modulation of a SOX21A-GATAe-mediated mechanism. Inhibition of *cad* expression in ISCs and EBs repress age-associated gut hyperplasia and promote healthy aging in *Drosophila*. This study not only elucidated a new function of the homeobox transcription factor *cad* in the regulation of stem/progenitor cell differentiation, but also suggests a potential approach for the promotion of healthy aging in humans.

## Results

### *Cad* expression increases in *Drosophila* intestinal stem and progenitor cells upon aging

To identify the potential regulators that modulate ISC aging, differentially expressed genes were analyzed in young and aged intestinal stem and progenitor cells. Analysis of published RNA-sequencing (RNA-seq) data [44] identified the homeobox transcription factor gene *caudal* (*cad*). This gene is highly expressed in ISCs and EBs in the R4 and R5 regions of the *Drosophila* midgut (S1B–S1D Fig). Real-time quantitative PCR (RT-qPCR) analyses using sorted *esg*-GFP$^+$ cells from young and old *Drosophila* showed that the *cad* mRNA is significantly upregulated in aged *esg*-GFP$^+$ cells (which indicate ISCs and their differentiating progenies) (S1E Fig).

As a homeobox transcription factor, *cad* upregulation has been reported in the posterior midgut of aged *Drosophila* [43], where it regulates gut immune homeostasis [41,42]. However, these studies did not show the cell-type specific expression pattern of *cad*. The function of

*cad* in intestinal stem/progenitor cells and gut epithelial homeostasis remains unknown. To characterize the cell-type-specific expression pattern of the endogenous CAD protein (encoded by *cad*) upon aging, the GFP-tagged protein trap line *cad-EGFP$^{VK00033}$* [45] was used. Consistent with the published RNA-seq data (S1D Fig), CAD protein expressed almost in all cell types of midguts (Fig 1A–1L). Immunostaining results showed that CAD mainly expresses in the posterior (R4 and R5 regions) midgut (S1G–S1I Fig) but barely or slightly expresses in the anterior (R1 and R2 regions) midguts and CCR (R3 region) midguts (S1G–S1I Fig). Observing the CAD-EGFP reporter line showed that the protein expression of CAD had significantly increased in intestinal epithelial cells in the R4 and R5 regions of the midgut upon aging (Fig 1A–1L). In addition, the expression of CAD in some ECs and EEs seemed higher than in ISCs of young *Drosophila* (Fig 1C, 1E and 1G). However, the increase of CAD proteins in ISCs and EBs (Fig 1A–1L) of old *Drosophila* was more dramatic compared with the increase in ECs or EEs (Fig 1A–1L). The western blot analyses also showed that the expression level of CAD dramatically increased in aged midguts compared with its expression in young midguts (S1F Fig). By co-staining of *esg*-lacZ, Dl, and CAD-EGFP, we found that the EBs (*esg*-lacZ$^+$Dl$^-$ cells) displayed a higher CAD expression level than the ISCs in the same ISC-EB pairs in young flies (Fig 1M and 1N). Moreover, paraquat (PQ)- or dextran sulphate sodium (DSS)-induced damages to *Drosophila* midgut could dramatically activate the expression of CAD (S1J–S1K Fig). The obtained results suggest that, except for regulating the gut immune response, *cad* might also function in ISCs or EBs and regulate gut homeostasis upon aging.

## Depletion of *cad* in intestinal stem and progenitor cells causes accumulation of premature enterocytes in the *Drosophila* midgut

To investigate the function of *cad* in *Drosophila* intestinal stem and progenitor cells, *cad* was depleted in ISCs and EBs via conditional temperature-sensitive *esg-Gal4* (*esg$^{ts}$*)-driven two different RNA interference (RNAi) lines (BDSC# 57546 and BDSC# 34702). The RT-qPCR analyses showed a significant reduction of *cad* mRNA expression in the midguts of flies carrying either *tub-Gal4>UAS-cad RNAi$^{BL57546}$* or *tub-Gal4>UAS-cad RNAi$^{BL34702}$* compared to that in the midguts of flies carrying *tub-Gal4>UAS-GFP* (S2A Fig).

Depletion of *cad* in ISCs and EBs promoted the progenitors (*esg$^+$* cells) to differentiate into ECs (Pdm1 positive and polyploid cells), as indicated by the increase of both *esg$^+$* and Pdm1$^+$ transient premature ECs (pre-ECs) in midguts (Figs 2A–2D and S2B–S2D). This phenotype is quite similar to that reported in *Drosophila* carrying *esg$^{ts}$*-driven *Sox21a* overexpression (a *Drosophila* Sox transcription factor that drives EB-to-EC differentiation [46–48]). Similar to the phenotype of overexpression of *Sox21a* in *esg$^+$* cells [48], we found that knockdown of *cad* in *esg$^+$* cells led to an increase in the number of cells in the *esg$^+$* clusters (Fig 2E). Besides, we found that depletion of *cad* in the EBs by *NRE-Gal4* (*NRE$^{ts}$*) also formed NRE$^+$ clusters (Fig 2F). Moreover, depletion of *cad* in *esg$^+$* cells did not decrease the number of *esg$^+$* cells but decreased the number of ISCs (Delta (Dl)$^+$ cells) (Figs 2G–2K and S2E–S2H) in the posterior region (PMG) of the midguts. Terminal deoxynucleotidyl transferase dUTP nick end labeling (TUNEL) analyses were used to detect cell death in the midguts. No significant increase in cell death was found in the midguts of *Drosophila* carrying *esg$^{ts}$*-driven *cad* RNAi compared with the controls (S2M and S2N Fig). Inhibition of cell apoptosis by depletion of *reaper* (*rpr*) in *esg$^+$* cells could not rescue the phenotype of Dl$^+$ cell loss caused by *cad* depletion (S2O–S2R Fig). Thus, we ruled out the possibility that the reduction of ISCs in *cad*-depleted flies was caused by ISC cell death. Moreover, we found that depletion of *cad* in *esg$^+$* cells also did not significantly affect the proliferation rate of ISCs (indicated by phosphor-histone H3 (pH3) staining) either under homeostatic conditions or after injury (Fig 2L and 2M).

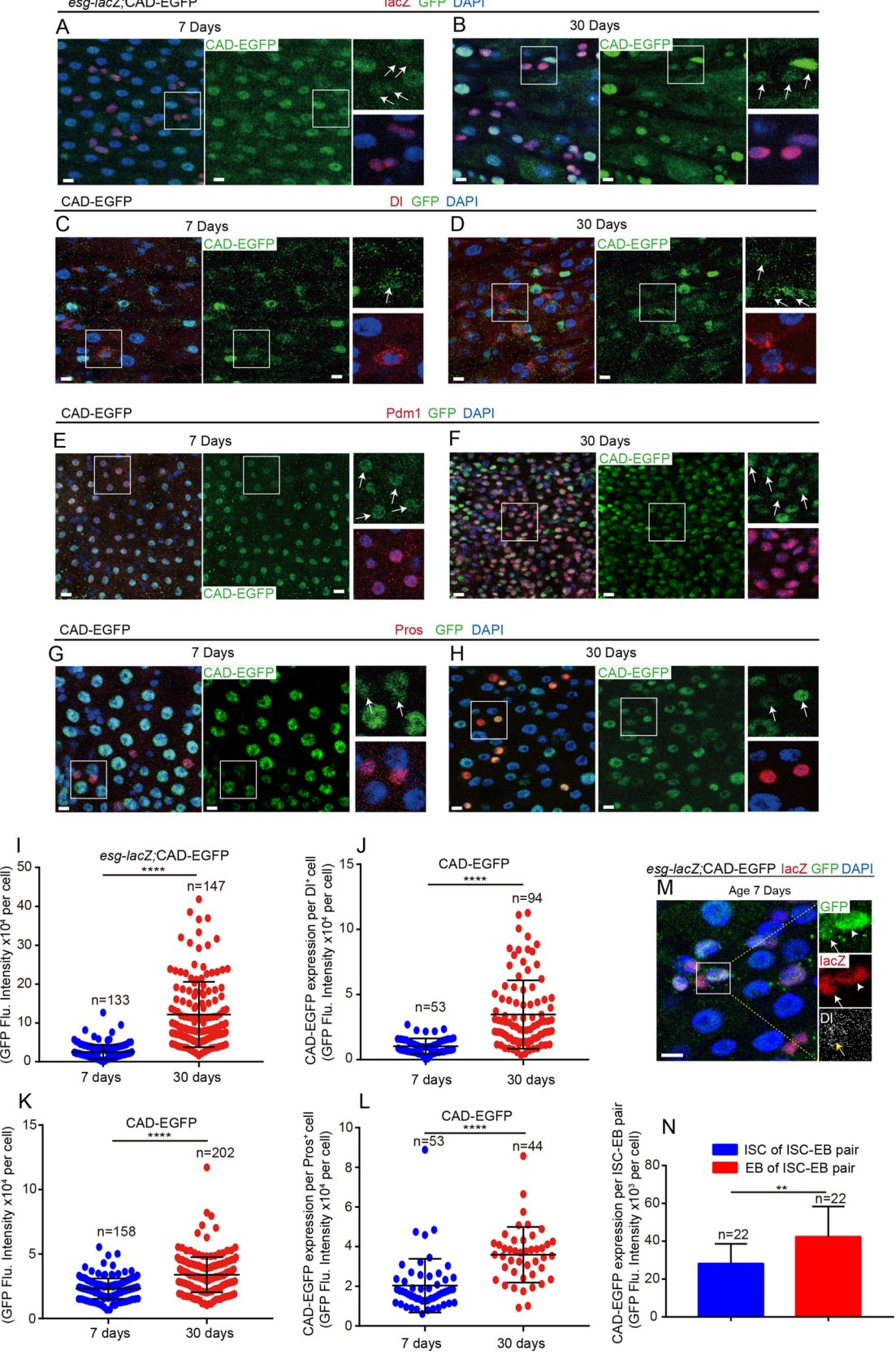

**Fig 1. Expression of *cad* dramatically increases in *Drosophila* intestinal stem cells (ISCs) and progenitor cells upon aging.** (A-B) Expression of endogenous CAD-EGFP protein (green) in *esg*[+] cells (ISCs and EBs; co-stained with *esg*-LacZ, red) in midguts from the R4 region of 7-day (A) and 30-day-old (B) *Drosophila*. The enlarged insets show *esg*-LacZ[+] cells (red) with CAD-EGFP (green) staining. Arrows indicate *esg*-LacZ[+] cells. (C-D) Expression of endogenous CAD-EGFP protein (green) in ISCs (labeled by Dl staining, red) in midguts from the R4 region of 7-day (C) and 30-day-old (D) *Drosophila*. The enlarged insets show ISCs (red) with CAD-EGFP (green) staining. Arrows indicate Dl[+] ISCs. (E-F) Expression of endogenous CAD-EGFP protein (green) in ECs (labeled by Pdm1 staining, red) in midguts from the R4 region of 7-day (E) and 30-day-old (F) *Drosophila*. The enlarged insets show ECs (red) with CAD-EGFP (green) staining. Arrows indicate Pdm1[+] ECs. (G-H) Expression of endogenous CAD-EGFP protein (green) in EEs (labeled by Prospero staining, red) in midguts from the R4 region of 7-day (G) and 30-day-old (H) *Drosophila*. The enlarged insets show EEs (red) with CAD-EGFP (green) staining. Arrows indicate Pros[+] EEs. (I) Quantification of fluorescence intensity of CAD-EGFP in *esg*-LacZ[+] cells of 7-day and 30-day-old *Drosophila* as shown in (A-B). (J) Quantification of fluorescence intensity of CAD-EGFP in Dl[+] ISCs of 7-day and 30-day-old *Drosophila* as shown in (C-D). (K) Quantification of fluorescence intensity of CAD-EGFP in Pdm1[+] ECs of 7-day and 30-day-old *Drosophila* as shown in (E-F). (L) Quantification of fluorescence intensity of CAD-EGFP in Pros[+] EEs of 7-day and 30-day-old *Drosophila* as shown in (G-H). (M) Representative images showing the expression pattern of CAD in young *Drosophila* midgut. The enlarged insets show *esg*-LacZ[+] cells (red) with CAD-GFP (green) staining. Arrows indicate *esg*-LacZ[+] Dl[+] cells (ISCs). Arrowheads indicate *esg*-LacZ[+] Dl[-] cells (EBs). (N) Quantification of fluorescence intensity of CAD-EGFP in ISCs or EBs from per ISC-EB pair as shown in (M). The number n is indicated. Each dot represents one cell. DAPI stained nuclei (blue). Scale bars represent 10μm (in Fig 1A-1H and 1M). Error bars represent SD. Significance was assessed via student's t-tests: $^*p < 0.05$, $^{**}p < 0.01$, $^{***}p < 0.001$, $^{****}p < 0.0001$, and ns (non-significant) represents $p > 0.05$. See also S1 Fig.

## Overexpression of *cad* in enteroblasts prevents enterocyte formation

Since depletion of *cad* promotes ISCs to differentiate into ECs, this study hypothesized that overexpression of this protein might prevent the progenitor cells from differentiating into mature ECs. Under normal conditions, the majority of ISCs in young *Drosophila* midguts are in a quiescent state [10,12,49,50]. As expected, under normal conditions overexpression of *cad* in *esg*[+] or in *NRE*[+] cells did not induce dramatic proliferation or differentiation phenotype (Figs 2L, S2E–S2H, and S3A–S3F). To induce these quiescent ISCs to become active and produce ECs, paraquat (PQ) was used to induce epithelial injury in the midgut [12]. After the damage had been induced, the ISCs in the control flies were activated and began to quickly proliferate and differentiate into ECs. During gut epithelial repair (after 24 h of PQ treatment and 24 h of regeneration), many transient pre-ECs (both *esg*[+] and Pdm1[+] cells) were found in the midguts of the control *Drosophila* carrying *UAS-GFP* (Fig 3A and 3C). However, in the midguts of *Drosophila* carrying *esg*[ts]-driven *cad* overexpression only diploid *esg*[+] cells (either ISCs or EBs), but few differentiating pre-ECs (both *esg*[+] and Pdm1[+] polyploid cells), were found (Fig 3B and 3C).

Moreover, we found that overexpression of *cad* in *esg*[+] cells led to a slight increase of ISC proliferation (indicated by the increases of pH3[+] cells, Dl[+] cells, and total *esg*[+] cells) after PQ injury (Figs 2M and S2I–S2L). The slight increase of ISC proliferation in *cad*-overexpressed in the midguts may because that overexpression of CAD in *esg*[+] cells caused the failure of ISC-to-EC differentiation, which in turn caused the formation of EB tumors in midguts. The EB tumors send EGF (Spi) and JAK-STAT ligands (Upd2) to neighboring ISCs to drive their proliferation [46,47].

To further delineate the role of *cad* in progenitor differentiation, *cad* was either specifically overexpressed or depleted in EBs using the temperature-sensitive NRE-*Gal4* (*NRE*[ts]) [51] or in ISCs using the temperature-sensitive *ISC-Gal4* (*ISC*[ts]) [51]. Depletion of *cad* in EBs (but not in ISCs) induced the accumulation of pre-ECs in the midgut (Figs 3D–3F and S3G–S3I). After injury (induced by PQ treatment), overexpression of *cad* in EBs (but not ISCs) led to the failure of the formation of pre-ECs (Figs 3G–3I and S3J–S3L). Thus, the regulation of *cad* expression in EBs is critical for regulating the ISC-to-EC differentiation and maintaining gut epithelial homeostasis.

To further demonstrate that *cad* participates in the promotion of ISC-to-EC differentiation, the lineages of *cad*-null (*cad*[2]) and *cad*-overexpressed ISCs were traced by performing mosaic

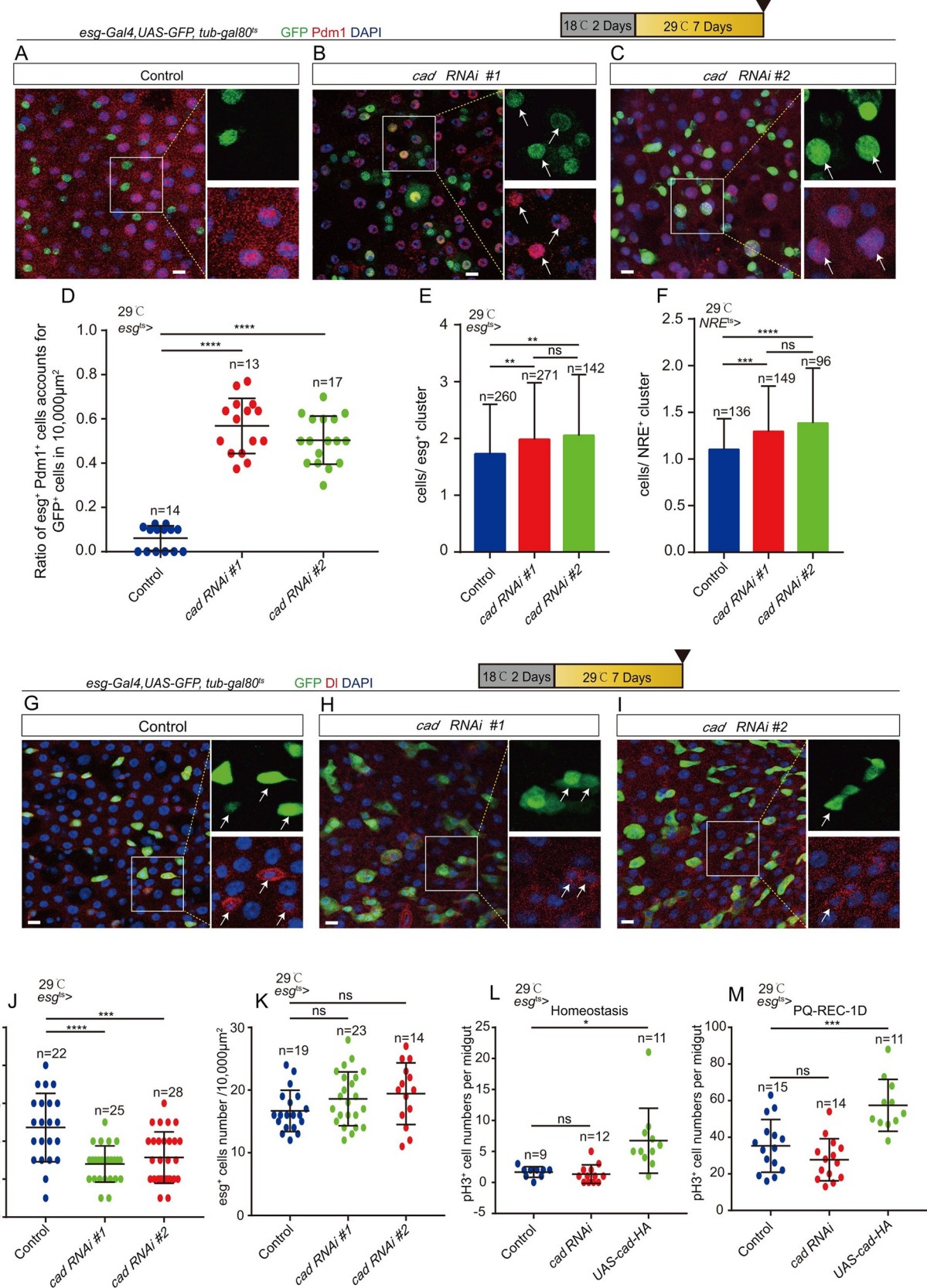

**Fig 2. Depletion of *cad* in ISCs and progenitor cells promotes ISC-to-EC differentiation.** (A-C) Immunofluorescence images of *esg*-GFP (green) and Pdm1 (red) staining with the midgut section from the R4 region of control *Drosophila* (A, *esg^ts*-Gal4>*UAS-GFP*) and the *cad*-depleted *Drosophila* by *esg^ts*-Gal4-driven two different *cad* RNAi lines (B, *cad* RNAi #1: BDSC #57546; C, *cad* RNAi #2: BDSC #34702). *esg*-GFP (green) represents ISCs and their differentiating cells. Pdm1 staining (red) was used to visualize differentiating ECs. White arrows indicate differentiating pre-ECs (both *esg*-GFP⁺ and Pdm1⁺ cells). *esg*-GFP⁺ and Pdm1⁻ cells are ISCs or EBs. *esg*-GFP⁻ and Pdm1⁺ cells are mature ECs. (D) Quantification of the ratio of *esg*-GFP⁺ and Pdm1⁺ cells account for GFP⁺ cells per 10,000 μm² area of the R4 region midguts as shown in (A-C). The number n represents the ROI in midguts from each experiment. One dot corresponds to one ROI (10,000 μm² area). (E) Quantification of the average number of cells in each *esg*-GFP⁺ cluster in midguts of control flies (*esg^ts*>*UAS-GFP*) and flies knocking down of *cad* in *esg*-GFP⁺ cells. (F) Quantification of the average number of cells in each *NRE*-GFP⁺ cluster in midguts of control flies (*NRE^ts*>*UAS-GFP*) and flies knocking down of *cad* in *NRE*-GFP⁺ cells. (G-I) Immunofluorescence images of *esg*-GFP (green) and Dl (red) staining with the midgut section from the R4 region of control *Drosophila* (G, *esg^ts*-Gal4>*UAS-GFP*) and *cad*-depleted *Drosophila* by *esg^ts*-Gal4-driven two different *cad* RNAi lines (H, *cad* RNAi #1; I, *cad* RNAi #2). *esg*-GFP (green) represents ISCs and their differentiating cells. Dl staining (red) was used to visualize ISCs. White arrows indicate Dl⁺ ISCs. (J-K) Quantification of the numbers of Dl⁺ ISCs (J) and *esg*-GFP⁺ cells (K) in a 10,000 μm² area of the R4 region midguts from control *Drosophila* and *Drosophila* carrying *esg^ts*-Gal4>*cad* RNAi as shown in (G-I). The number n represents the ROI in midguts from each experiment. One dot corresponds to one ROI (10,000 μm² area). (L) Quantification of the pH3⁺ number in the whole midgut from control flies (*esg^ts*-Gal4-driven *UAS-GFP*), *Drosophila* carrying *esg^ts*-Gal4-driven *cad* RNAi, and *Drosophila* carrying *esg^ts*-Gal4-driven *UAS-cad-HA* in homeostasis. The number n is indicated. Each dot represents one midgut. (M) Quantification of the pH3⁺ number in the whole midgut from control flies (*esg^ts*-Gal4-driven *UAS-GFP*), *Drosophila* carrying *esg^ts*-Gal4-driven *cad* RNAi, and *Drosophila* carrying *esg^ts*-Gal4-driven *UAS-cad-HA*. Flies were treated with PQ for one day, then recovered on normal food for 1 day (hereafter referred to as PQ-REC-1D). The number n is indicated. Each dot represents one midgut. DAPI stained nuclei (blue). Scale bars represent 10 μm (Fig 2A–2C and 2G–2I). Error bars represent SD. Student's t-tests were used to assess statistical significane as *$p < 0.05$, **$p < 0.01$, ***$p < 0.001$, ****$p < 0.0001$, while ns (non-significant) represents $p > 0.05$. See also S2 Fig.

analysis with repressible cell marker (MARCM) [5,6]. After clone induction (ACI) for 7 days, wild-type clones contained a few Pdm1 negative (Pdm1⁻) diploid cells and a cluster of Pdm1⁺ polyploid ECs (Fig 3J); however, *cad*-overexpressing clones almost exclusively consisted of Pdm1 negative (Pdm1⁻) diploid cells (few Pdm1⁺ polyploid ECs were found in *cad*-overexpressing clones) (Fig 3L). By calculating the ratio of Pdm1⁺ polyploid ECs per clone, the *cad*-null clones (Fig 3K) were found to show an obvious increase in EC production, while *cad*-overexpressing clones showed a significant failure of EC production compared with the wild-type control (*FRT40A*) (Fig 3J and 3M). Interestingly, *cad*-overexpressing clones showed a significantly decreased clone size compared with the wild-type control (*FRT40A*) (Fig 3N). This proliferation defect of *cad*-overexpressing clones was similar to the *Sox21a* mutant clones [46]. Moreover, we detected the composition of cell types in *cad*-null and *cad*-overexpressing clones (Figs 3O–3R and S3M–S3O). The results showed that *cad*-overexpressing and *cad* mutant clones exhibited a significant low ratio of Dl⁺ cells compared to genotype control clones (*FRT40A*) (Fig 3O and 3R). These cells in *cad*-overexpression clones did not express the EE marker Prospero (S3O Fig). In addition, the average size of Pdm1⁺ ECs in the *Caudal*-null clones exceeded that of the control clones (*FRT40A*) (Fig 3S). This suggests that *cad* might regulate the endopolyploidy process of ISC-to-EC differentiation.

## *Cad* regulates intestinal stem cell to enterocyte differentiation by modulating *Sox21a* expression

To investigate the mechanism with which *cad* regulates ISC-to-EC differentiation, RNA-seq was performed on dissected midguts with *cad* depleted in *esg*⁺ cells (*esg^ts*> *cad RNAi*) as well as control midguts carrying *esg^ts*-Gal4 only (S1 Table).

A number of genes that have been reported to regulate the differentiation of ISCs or progenitor cells in *Drosophila* were significantly upregulated in midguts with *cad* depleted in *esg*⁺ cells (Fig 4A). Among these genes, the SOX family transcription factor SOX21A, which is essential for ISC-to-EC differentiation, was found [46–48]. RT-qPCR analyses using sorted *esg*-GFP⁺ cells also showed that the mRNA expression of *Sox21a* and several demonstrated downstream genes of *Sox21a* (including *pdm1*, *GATAe*, *Connectin*, *armadillo*, and *E-cadherin*) significantly increased in the midgut of *cad*-depleted *Drosophila* (Fig 4B). Since *cad* mutant

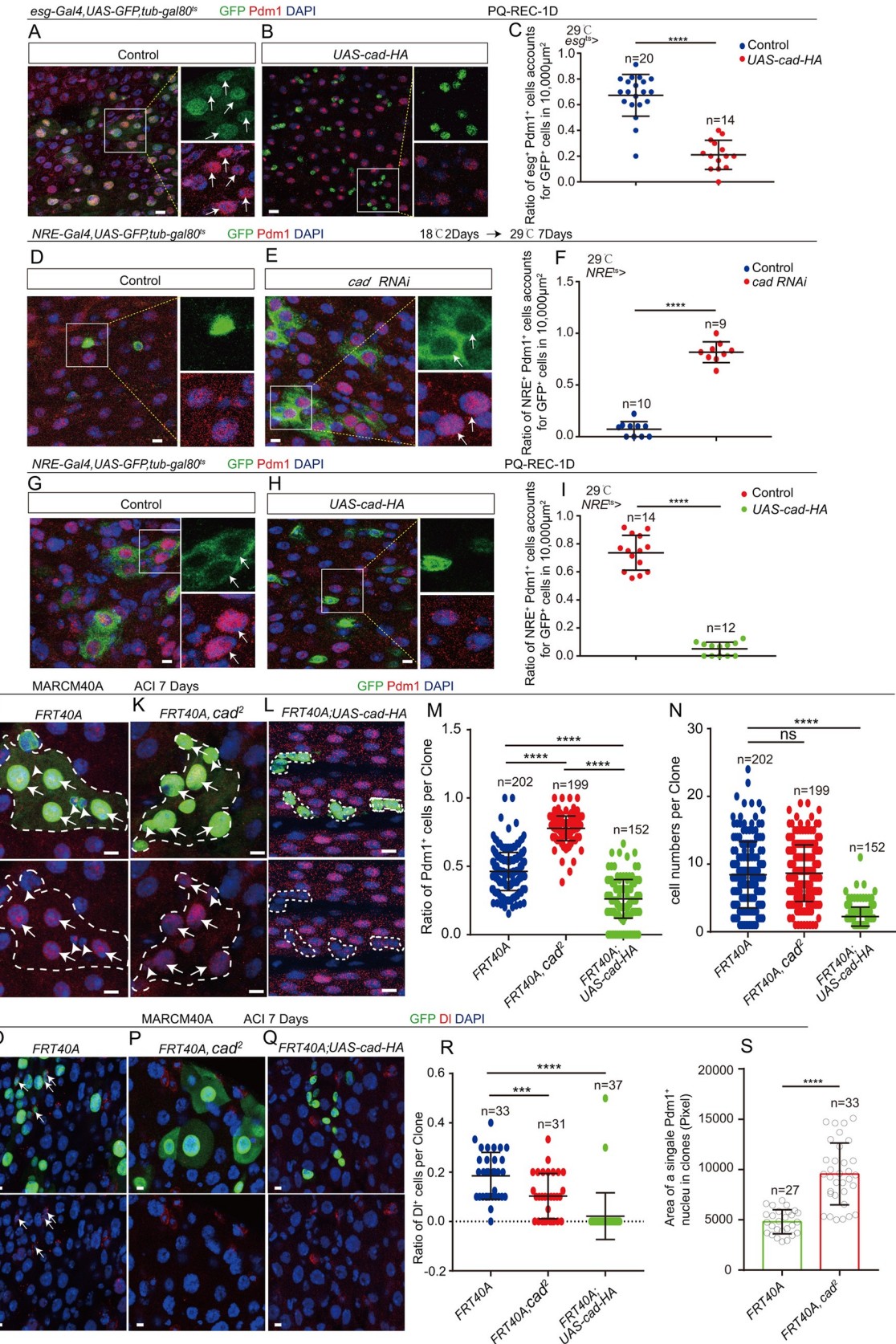

**Fig 3. Overexpression of cad in EBs prevents ISCs to produce differentiated ECs.** (A-B) Immunofluorescence images of *esg*-GFP (green) and Pdm1 (red) staining with the midgut section from the R4 region of control *Drosophila* (A, *esg*$^{ts}$*-Gal4>UAS-GFP*) and *Drosophila* carrying *esg*$^{ts}$*-Gal4> UAS-cad-HA* (B). *Drosophila* were treated with PQ (PQ-REC-1D). *esg*-GFP (green) represents ISCs and their differentiating cells. Pdm1 staining (red) was used to visualize differentiating ECs. White arrows indicate differentiating pre-ECs (*esg*-GFP$^+$ and Pdm1$^+$ cells). *esg*-GFP$^+$ and Pdm1$^-$ cells are ISCs or EBs. *esg*-GFP$^-$ and Pdm1$^+$ cells are mature ECs. (C) Quantification of the ratio of *esg*-GFP$^+$ and Pdm1$^+$ cells in a 10,000 μm$^2$ area of the R4 region of midguts of *Drosophila* as shown in (A-B). The number n represents the ROI in midguts from each experiment. One dot corresponds to one ROI (10,000 μm$^2$ area). (D-E) Immunofluorescence images of *NRE*-GFP (green) and Pdm1 (red) staining with the midgut section from the R4 region of 9-day-old (2 days at 18˚C, then 7 days at 29˚C) control *Drosophila* (D, *NRE*$^{ts}$*-Gal4>UAS-GFP*) and *cad*-depleted *Drosophila* carrying *NRE*$^{ts}$*-Gal4> UAS-cad RNAi* (E). *NRE*-GFP (green) represents EBs. Pdm1 staining (red) was used to visualize differentiating ECs. White arrows indicate differentiating pre-ECs (*NRE*-GFP$^+$ and Pdm1$^+$ cells). *NRE*-GFP$^+$ and Pdm1$^-$ cells are EBs. *NRE*-GFP$^-$ and Pdm1$^+$ cells are mature ECs. (F) Quantification of the ratio of *NRE*-GFP$^+$ and Pdm1$^+$ cells in a 10,000 μm$^2$ area of the R4 region of midguts as shown in (D-E). The number n represents the ROI in midguts from each experiment. One dot corresponds to one ROI (10,000 μm$^2$ area). (G-H) Immunofluorescence images of *NRE*-GFP (green) and Pdm1 (red) staining with the midgut section from the R4 region of control *Drosophila* (G, *NRE*$^{ts}$*-Gal4>UAS-GFP*) and *Drosophila* carrying *NRE*$^{ts}$*-Gal4>UAS-cad-HA* (H). *Drosophila* were treated with PQ (PQ-REC-1D). *NRE*-GFP (green) represents EBs. Pdm1 staining (red) was used to visualize differentiating ECs. White arrows indicate differentiating pre-ECs (*NRE*-GFP$^+$ and Pdm1$^+$ cells). *NRE*-GFP$^+$ and Pdm1$^-$ cells are EBs. *NRE*-GFP$^-$ and Pdm1$^+$ cells are mature ECs. (I) Quantification of the ratio of *NRE*-GFP$^+$ and Pdm1$^+$ cells account for GFP$^+$ cells in a 10,000 μm$^2$ area of the R4 region midguts as shown in (G-H). The number n represents the ROI in midguts from each experiment. One dot corresponds to one ROI (10,000 μm$^2$ area). (J-L) Immunofluorescence analyses of control (*FRT40A*, J), *cad*$^2$ mutant (K), and *cad-HA* overexpressing (L) mosaic analysis with repressible cell marker (MARCM) clones (green, outlined by white dotted lines) 7 days after clone induction (ACI) of flies. Pdm1 staining (red) was used to visualize ECs. White arrows indicate Pdm1$^+$ polyploid ECs, and white arrowheads indicate Pdm1$^-$ diploid cells in (J). (M) Quantification of the ratio of Pdm1$^+$ cells per clone with indicated genotypes of experiments presented in (J-L). Each dot corresponds to one clone. (N) Quantification of the GFP$^+$ cells per clone with indicated genotypes of experiments presented in (J-L). Each dot corresponds to one clone. (O-Q) Immunofluorescence analyses of control (*FRT40A*, O), *cad*$^2$ mutant (P), and *cad-HA* overexpressing (Q) mosaic analysis with repressible cell marker (MARCM) clones (green) 7 days after clone induction (ACI) of flies. Dl staining (red) was used to visualize ISCs. White arrows indicate ISCs in clones. (R) Quantification of the ratio of Dl$^+$ cells per clone with indicated genotypes of experiments presented in (O-Q). Each dot corresponds to one clone. (S) Quantification of the average size of Pdm1$^+$ ECs in control clones (*FRT40A*) and *cad*$^2$ mutant clones. The number n represents Pdm1$^+$ ECs in clones. DAPI-stained nuclei (blue). Scale bars represent 10μm (Fig 3A, 3B, 3D, 3E, 3G and 3H), or 5μm (Fig 3J–3L and 3O–3Q). Error bars represent SD. Student's t-tests were used to assess statistical significance: $^*p < 0.05$, $^{**}p < 0.01$, $^{***}p < 0.001$, $^{****}p < 0.0001$, and ns (non-significant), which represents $p > 0.05$. See also S3 Fig.

has a similar differentiation defect to SOX21A overexpression in EBs [46], it is reasonable to hypothesize that *cad* might regulate the ISC-to-EC differentiation through a SOX21A-mediated mechanism.

To further confirm the results of the RNA-seq analyses (i.e., that *cad* regulates the expression of *Sox21a* in *esg*$^+$ cells), the endogenous SOX21A reporter line *Sox21a-HA* [52] was used. As expected, the expression of SOX21A-HA protein in *cad*-depleted *esg*$^+$ cells was stronger compared with its expression in the control *esg*$^+$ cells (Fig 4C–4E). Moreover, the western blot analyses also showed that the expression level of SOX21A-HA dramatically increased in *cad*-depleted midguts compared with its expression in the control midguts (S4A–S4C Fig). Interestingly, we noticed that, while the expression of CAD in the intestinal stem/progenitor cells increased upon aging, the expression of SOX21A in these stem/progenitor cells declined (S4D, S4E, S4G, S4H and S4I Fig).

After injury, while SOX21A expression was dramatically upregulated in the control *esg*$^+$ cells (Figs 4F, 4H and S4F), it still retained a relatively low level in *cad*-overexpressing *esg*$^+$ cells (Fig 4G and 4H). More importantly, forced expression of *Sox21a* in *esg*$^+$ cells fully restored the failure of *cad*-overexpressed *esg*$^+$ cells to produce pre-ECs after injury as evidenced by the generation of both Pdm1$^+$ and *esg*$^+$ cells (Fig 4I–4M). These data suggest that *cad* regulates ISC-to-EC differentiation by inhibiting the expression of *Sox21a* in *esg*$^+$ cells.

## *GATAe* functions as a downstream of *cad* to regulate intestinal stem cell to enterocyte differentiation

Since previous studies have shown that *GATAe* (a gene encoding a GATA family transcription factor, which is a downstream of *Sox21a*) regulates ISC-to-EC differentiation in *Drosophila*

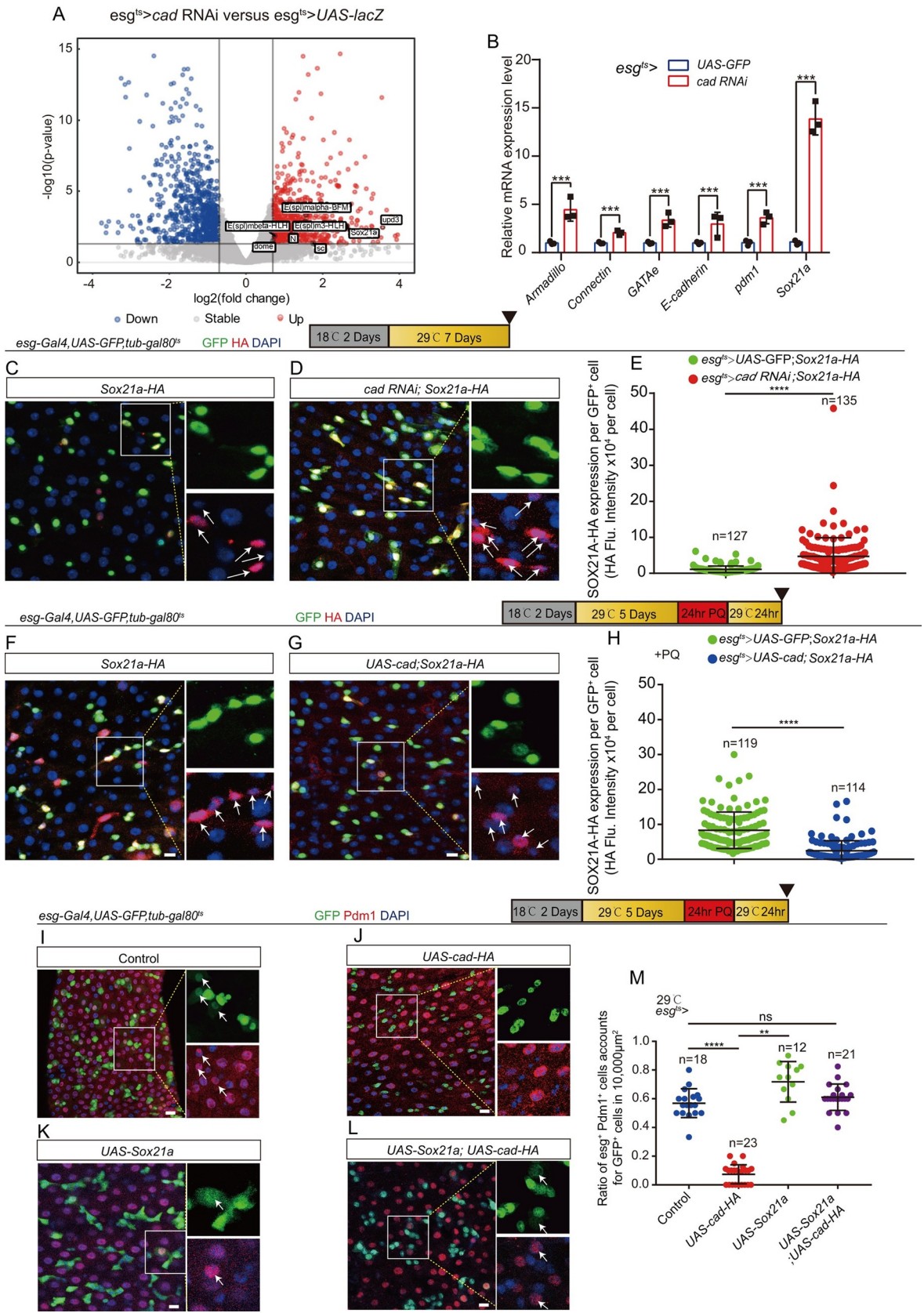

**Fig 4. *Cad* regulates ISC-to-EC differentiation by modulating *Sox21a* expression.** (A) Volcano plots of differentially expressed genes in a pair-wise comparison of *cad*-depleted *Drosophila* (*esg^ts*-Gal4>*cad* RNAi) midguts to control *Drosophila* (*esg^ts*-Gal4>*UAS- lacZ*) midguts. (B) Relative mRNA fold changes of *Armadillo*, *Connectin*, *GATAe*, *E-cadherin*, *pdm1*, and *Sox21a* in sorted *esg^+* cells from of young (2 days at 18˚C then transferred to 29˚C for 5 days) genotype control flies' midguts (blue lines, *esg^ts*-Gal4-driven *UAS-GFP*) and *cad*-depleted midguts (red lines, *esg^ts*-Gal4-driven *cad* RNAi). The changes of expressions were plotted relative to genotype controls, which was set to 1. Error bars indicate the standard deviation (SD) of three independent experiments. (C-D) Representative images showing the expression of endogenous SOX21A-HA (red) in midguts of *Drosophila* carrying *esg^ts*-Gal4>*UAS-GFP* (control; C) and midguts carrying *esg^ts*-Gal4>*cad* *RNAi* (D). The right enlarged insets from (C-D) show the single channel of *esg*-GFP (green) and SOX21A-HA (red). (E) Quantification of fluorescence intensity of SOX21A-HA with indicated genotypes as shown in (C-D). One dot represents one *esg*-GFP^+ cell. (F-G) Representative images showing the expression of endogenous SOX21A-HA (red) in midguts of *Drosophila* carrying *esg^ts*-Gal4>*UAS-GFP* (control; F) and midguts carrying *esg^ts*-Gal4>*UAS-cad* (G). *Drosophila* were treated with PQ (PQ-REC-1D). The right enlarged insets from (F-G) show the single channel of esg-GFP (green) and SOX21A-HA (red). (H) Quantification of fluorescence intensity of SOX21A-HA with indicated genotypes as shown in (F-G). One dot represents one *esg*-GFP^+ cell. (I-L) Immunofluorescence images of *esg*-GFP (green) and Pdm1 (red) staining with the midgut section from the R4 region of control *Drosophila* (I; *esg^ts*-Gal4>*UAS-GFP*), *Drosophila* carrying *esg^ts*-Gal4>*UAS-cad-HA* (J), *Drosophila* carrying *esg^ts*-Gal4>*UAS-Sox21a* (K), and *Drosophila* carrying *esg^ts*-Gal4-driven *UAS-cad-HA* and *UAS-Sox21a* (L). *Drosophila* were treated with PQ. *esg*-GFP (green) represents ISCs and their differentiating cells. Pdm1 staining (red) was used to visualize differentiating ECs. White arrows indicate differentiating pre-ECs (*esg*-GFP^+ and Pdm1^+ cells). *esg*-GFP^+ and Pdm1^- cells are ISCs or EBs. *esg*-GFP^- and Pdm1^+ cells are mature ECs. (M) Quantification of the ratio of *esg*-GFP^+ and Pdm1^+ cells per 10,000 μm² area of the R4 region midguts as indicated in (I-L). The number n represents the ROI in midguts from each experiment. One dot corresponds to one ROI (10,000 μm² area). DAPI stained nuclei (blue). Scale bars represent 10 μm (Fig 4C, 4D, 4F, 4G and 4I–4L). Error bars represent SD. Student's t-tests were used to assess significance: *$p < 0.05$, **$p < 0.01$, ***$p < 0.001$, ****$p < 0.0001$, and ns (non-significant), which represents $p > 0.05$. See also S4 Fig.

[48,52], the genetic interaction between *cad* and *GATAe* was tested. Depletion of *GATAe* in either *esg^+* cells (Fig 5A–5E) or EBs (Fig 5F–5J) significantly restored the differentiation defect of the accumulation of pre-ECs seen in *cad*-depleted midguts as evidenced by decreases of both *esg^+* and Pdm1^+ cells. Overexpression of *cad* in *esg^+* cells did not show a noticeable rescue of the defect of the accumulation pre-ECs seen in *GATAe*-overexpressing midguts (S5A–S5E Fig). Thus, *GATAe* functioned as a downstream gene of *cad* to regulate ISC-to-EC differentiation in *Drosophila*.

Since it has been shown that *GATAe* regulates ISC-to-EC differentiation by modulating the Notch signaling [53], the genetic interaction between *cad* and the Notch signaling pathway was also tested. Depletion of Notch in *esg^+* cells significantly restored the increase of pre-ECs production in midguts with *cad*-depleted *esg^+* cells (Fig 5K–5O).

## *Cad* promotes intestinal stem cell to enterocyte differentiation through manipulation of the JAK/STAT signaling pathway

Previous studies have shown that the Janus kinase/signal transducers and activators of transcription (JAK/STAT) signaling pathway functions as a upstream of *Sox21a* in EBs and promotes the maturation of ECs [46,48]. Thus, the genetic interaction between *cad* and the JAK/STAT signaling pathway was explored. After injury (induced by PQ treatment), while the activity of JAK/STAT signaling was significantly upregulated in control midguts, as indicated by the 10XSTAT-GFP transcriptional reporter (Fig 6A and 6C), it still retained a relatively low level in *cad*-overexpressing midguts (Fig 6B and 6C).

More importantly, after gut injury, forced expression of a constitutively active version of JAK (i.e., *hopscotch* (*hop*)) (*UAS-hop^TUM*) in ISCs and EBs by *esg^ts*-Gal4, or only in EBs by *NRE^ts*-Gal4, significantly restored the ISC-to-EC differentiation defect of *cad*-overexpressing midguts. This was evidenced by the generation of both *esg^+* and Pdm1^+ cells (Figs 6D–6H and S6A–S6E). In addition, depletion of *Stat92E* in ISCs and EBs by *esg^ts*-Gal4 significantly rescued the defect of pre-EC accumulation of *cad*-depleted midguts (Fig 6I–6M). In summary, these findings show that *cad* participates in the regulation of ISC-to-EC differentiation through the JAK/STAT-SOX21A-GATAe cascade in intestinal progenitor cells.

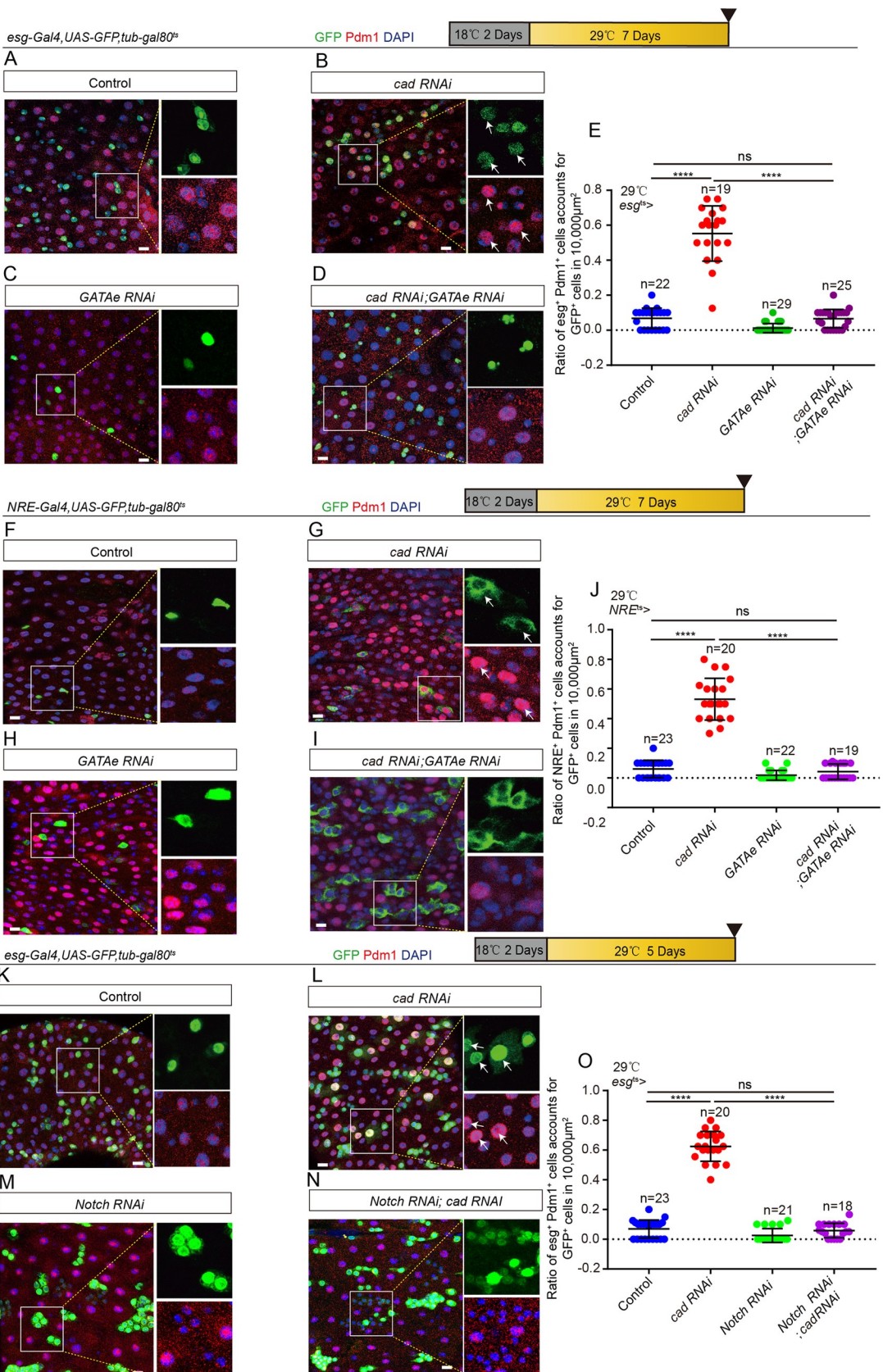

**Fig 5. *GATAe* functions downstream of *cad* to regulate ISC-to-EC differentiation.** (A-D) Immunofluorescence images of *esg*-GFP (green) and Pdm1 (red) staining with the midgut section from the R4 region of control *Drosophila* (A, *esg^{ts}*-*Gal4>UAS-GFP*), *Drosophila* carrying *esg^{ts}-Gal4>cad RNAi* (B), *Drosophila* carrying *esg^{ts}-Gal4>GATAe RNAi* (C), and *Drosophila* carrying *esg^{ts}-Gal4*-driven *cad RNAi* and *GATAe RNAi* (D), under normal conditions. *esg*-GFP (green) represents ISCs and their differentiating cells. Pdm1 staining (red) was used to visualize differentiating ECs. White arrows indicate differentiating pre-ECs (*esg*-GFP⁺ and Pdm1⁺cells). *esg*-GFP⁺ and Pdm1⁻ cells are ISCs or EBs. *esg*-GFP⁻ and Pdm1⁺ cells are mature ECs. (E) Quantification of the ratio of *esg*-GFP⁺ and Pdm1⁺ cells per 10,000 μm² area of the R4 region midguts of *Drosophila* with genotypes as indicated in (A-D). The number n represents the ROI in midguts from each experiment. One dot corresponds to one ROI (10,000 μm² area). (F-I) Immunofluorescence images of *NRE*-GFP (green) and Pdm1 (red) staining of the midgut section from the R4 region of control *Drosophila* (F, *NRE^{ts}-Gal4>UAS-GFP*), *Drosophila* carrying *NRE^{ts}-Gal4>cad RNAi* (G), *Drosophila* carrying *NRE^{ts}-Gal4>GATAe RNAi* (H), and *Drosophila* carrying *NRE^{ts}-Gal4*-driven *cad RNAi* and *GATAe RNAi* (I), under normal conditions. Pdm1 staining (red) visualizes differentiating ECs. White arrows indicate differentiating pre-ECs (*NRE*-GFP⁺ and Pdm1⁺ cells). *NRE*-GFP⁺ and Pdm1⁻ cells are EBs. *NRE*-GFP⁻ and Pdm1⁺ cells are mature ECs. (J) Quantification of the ratio of *NRE*-GFP⁺ and Pdm1⁺ cells per 10,000 μm² area of the R4 region midguts as indicated in (F-I). The number n represents the ROI in midguts from each experiment. One dot corresponds to one ROI (10,000 μm² area). (K-N) Immunofluorescence images of *esg*-GFP (green) and Pdm1 (red) staining with the midgut section from the R4 region of control *Drosophila* (K, *esg^{ts}-Gal4>UAS-GFP*), *Drosophila* carrying *esg^{ts}-Gal4>cad RNAi* (L), *Drosophila* carrying *esg^{ts}-Gal4>Notch RNAi* (M), and *Drosophila* carrying *esg^{ts}-Gal4*-driven *cad RNAi* and *Notch RNAi* (N), under normal conditions. *esg*-GFP (green) represents ISCs and their differentiating cells. Pdm1 staining (red) was used to visualize differentiating ECs. White arrows indicate differentiating pre-ECs (*esg*-GFP⁺ and Pdm1⁺cells). *esg*-GFP⁺ and Pdm1⁻ cells are ISCs or EBs. *esg*-GFP⁻ and Pdm1⁺ cells are mature ECs. (O) Quantification of the ratio of *esg*-GFP⁺ and Pdm1⁺ cells per 10,000 μm² area of the R4 region midguts as indicated in (K-N). The number n represents the ROI in midguts from each experiment. One dot corresponds to one ROI (10,000 μm² area). DAPI stained nuclei (blue). Scale bars represent 10 μm (Fig 5A–5D, 5F–5I and 5K–5N). Error bars represent SD. Student's t-tests were used to assess statistical significance: *p < 0.05, **p < 0.01, ***p < 0.001, ****p < 0.0001, and NS (non-significant), which represents p > 0.05. See also S5 Fig.

## Reduction of *cad* expression in intestinal stem and progenitor cells represses age-associated gut hyperplasia in *Drosophila*

The ISCs and progenitor cells in aged *Drosophila* midguts undergo malignant proliferation and show decreased differentiation efficiency [19,20]. These results in the continuous accumulation of ISCs and their differentiating progenies (*esg*⁺ polyploid cells) in aged midguts [19,20]. Previous studies have shown that the failure of ISC-to-EC differentiation causes the undifferentiated EBs sending EGF (Spi) and JAK/STAT ligands (Upd2) to neighboring ISCs to drive their proliferation [46,47]. In addition, the accumulation of undifferentiated EBs caused niche EC detachment [54] and up-regulating paracrine EGF (Vn) and JAK/STAT ligands (Upd3) expression further accelerated ISC proliferation. Thus, the age-associated gut hyperplasia of aged *Drosophila* might be increased by the age-associated decrease of the differentiation efficiency of ISCs. This study showed that *cad* is upregulated in ISCs and EBs in aged *Drosophila*, while a high level of *cad* expression in EBs repressed the differentiation of intestinal progenitor cells. Based on these findings, the following hypothesis was proposed: the age-associated upregulation of *cad* in ISCs and EBs contributes to the decrease of differentiation efficiency and gut hyperplasia in aged *Drosophila*.

To test this hypothesis, the expression of *cad* in ISCs and EBs was lowered by using *esg^{ts}*-driven *cad* RNAi. The differentiation capacities of ISCs and EBs appear to decrease with age, which leads to the accumulation of Esg and Dl expressing cells in the aged midgut [55]. Therefore, this study tested whether the reduction of CAD expression could repress aging-related *esg*⁺ cells, Dl⁺ cells and *esg*⁺Pdm1⁺ cells in the midguts of aged *Drosophila*. The midguts of 35-day-old *Drosophila* (*Drosophila* were cultured at 18˚C for 25 days and transferred to 29˚C for another 10 days to induce RNAi), which induced *esg^{ts}*-driven *cad* RNAi at middle age (i.e., the 25th day after eclosion), indeed showed a significantly lower level of hyperplasia compared with the midguts of control flies aged 35 days as indicated by less accumulation of *esg*⁺Pdm1⁺ cells (Fig 7A–7C), *esg*⁺ cells (Figs 7D–7F and S7A and S7B) and Dl⁺ cells (Fig 7I–7K). TUNEL analyses indicated that depletion of *cad* in aged *Drosophila* did not cause more cell death compared to the control (S7E and S7F Fig). We found that knockdown of CAD either in *esg*⁺ cells or EBs significantly reduced the ISC hyper-proliferation defect of aged flies as indicated by

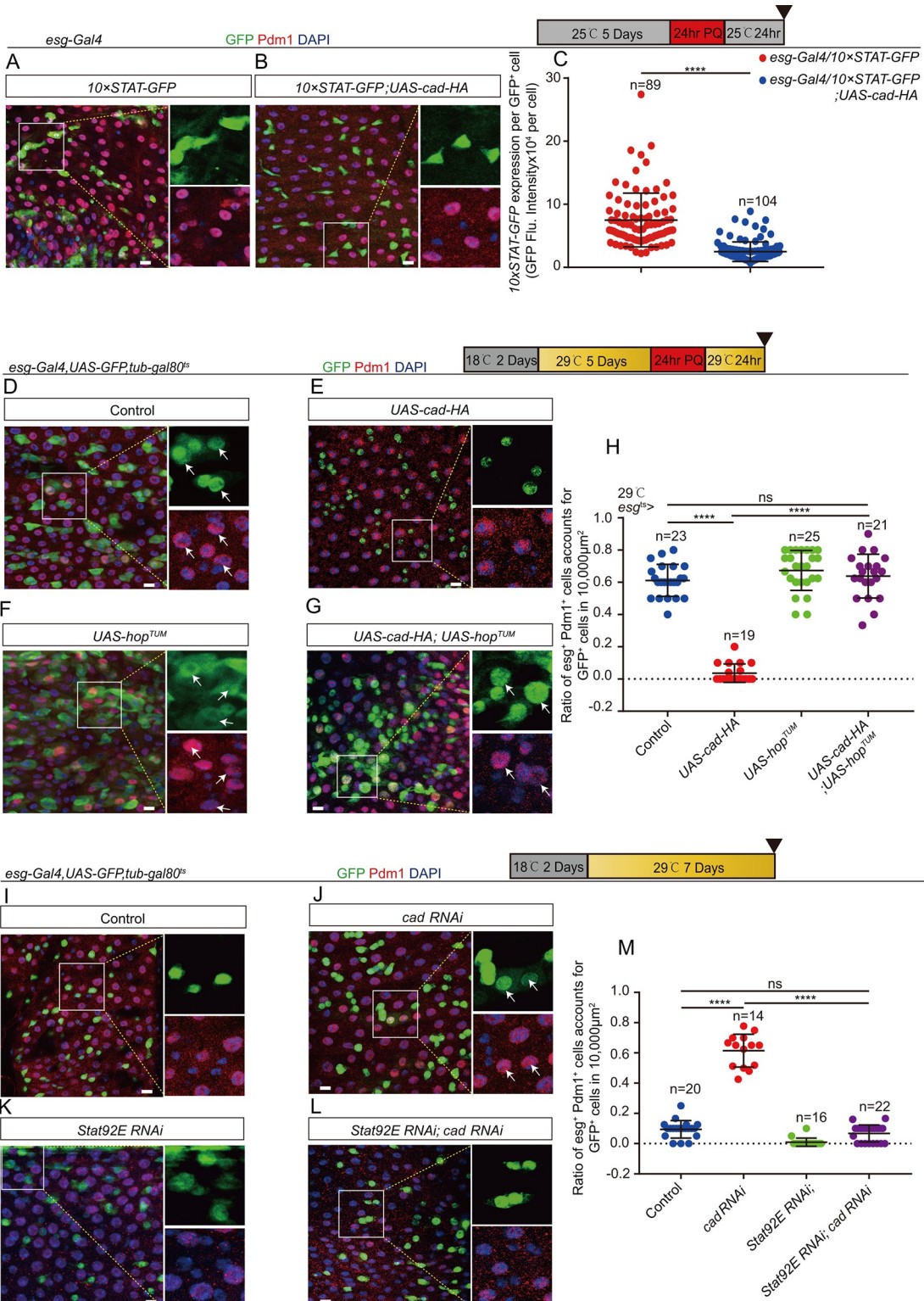

**Fig 6. *Cad* promotes ISC-to-EC differentiation by manipulating the JAK/STAT signaling pathway.** (A-B) Representative images showing the expression of 10xSTAT-GFP (green) in midguts of *Drosophila* carrying *esg-Gal4* only (control; A) and *Drosophila* carrying *esg-Gal4>UAS-cad-HA* (B) treated with PQ. Pdm1 (red) staining identifies pre-ECs and matured ECs. (C) Quantification of fluorescence intensity of 10x*STAT*-GFP per *esg*⁺ cells in midguts with genotypes as indicated in (A-B). The number n represents counted cells. (D-G) Immunofluorescence images of *esg*-GFP (green) and Pdm1 (red) staining with the

midgut section from the R4 region of control *Drosophila* (D, *esg^ts^-Gal4>UAS-GFP*), *Drosophila* carrying *esg^ts^-Gal4>UAS-cad-HA* (E), *Drosophila* carrying *esg^ts^-Gal4>UAS-hop^TUM^* (F), and *Drosophila* carrying *esg^ts^-Gal4*-driven *UAS-cad-HA* and *UAS-hop^TUM^* (G). *Drosophila* were treated with PQ. *esg*-GFP (green) identifies ISCs and their differentiating cells. Pdm1 staining (red) was used to visualize differentiating ECs. White arrows indicate differentiating pre-ECs (*esg*-GFP^+^ and Pdm1^+^cells). *esg*-GFP^+^ and Pdm1^-^ cells are ISCs or EBs. *esg*-GFP^-^ and Pdm1^+^ cells are matured ECs. (H) Quantification of the ratio of *esg*-GFP^+^ and Pdm1^+^ cells per 10,000 μm² area of the R4 region midguts as indicated in (D-G). The number n represents the ROI in midguts from each experiment. One dot corresponds to one ROI (10,000 μm² area). (I-L) Immunofluorescence images of *esg*-GFP (green) and Pdm1 (red) staining with the midgut under normal condition. The midgut section from the R4 region of control flies (I, *esg^ts^-Gal4*-driven *UAS-GFP*), *Drosophila* carrying *esg^ts^-Gal4*-driven *cad RNAi* (J), *Drosophila* carrying *esg^ts^-Gal4*-driven *Stat92E RNAi* (K), and *Drosophila* carrying *esg^ts^-Gal4*-driven *cad RNAi* and *Stat92E RNAi* (L). *esg*-GFP (green) identifies ISCs and their differentiating cells. Pdm1 staining (red) was used to visualize differentiating ECs. White arrows indicate differentiating pre-ECs (*esg*-GFP^+^ and Pdm1^+^cells). *esg*-GFP^+^ and Pdm1^-^ cells are ISCs or EBs. *esg*-GFP^-^ and Pdm1^+^ cells are mature ECs. (M) Quantification of the ratio of *esg*-GFP^+^ and Pdm1^+^ cells per 10,000 μm² area of the R4 region midguts as indicated in (I-L). The number n represents the region of interest in midguts from each experiment. One dot corresponds to one region of interest (ROI = 10,000 μm² area). DAPI stained nuclei are shown in blue. Scale bars represent 10 μm (Fig 6A, 6B, 6D–6G and 6I–6L). Error bars represent SD. Student's t-tests were used to assess statistical significance: *p < 0.05, **p < 0.01, ***p < 0.001, ****p < 0.0001, and NS (non-significant), which represents p > 0.05.

pH3^+^ staining (Fig 7G and 7H). Since overexpression of *cad* in *esg*^+^ cells of injured young *Drosophila* prevents ISC-to-EC differentiation, which in turn led to an increase in ISC proliferation and progenitor cell accumulation (Figs 2M and S2I–S2L), it is reasonable to predict that knockdown of cad in *esg*^+^ cells of aged *Drosophila* prevents gut hyperplasia by promoting intestinal progenitor cell differentiation.

As expected, knockdown of *cad* in *esg*^+^ cells of aged *Drosophila* promoted the activate of JAK/STAT signaling pathway as indicated by the upregulation of *Socs36E* (a demonstrated downstream of JAK/STAT pathway [56]) transcription (Fig 7L). Moreover, knockdown of either *Stat92E* or *Sox21a* restored the effect of *cad* RNAi-mediated reduction of gut hyperplasia of aged *Drosophila* (Fig 7M–7P).

In addition, overexpression of CAD in *esg*^+^ cells caused more severe gut hyperplasia defect in aged midgut indicated by pH3^+^ staining (S7C Fig) and *esg*-GFP staining (S7D Fig), and an increase of mortality of flies under low dose PQ-induced injury compared to the control flies (Fig 7Q). These data suggest that the upregulation of *cad* expression in ISCs and EBs in aged *Drosophila* might be an important cause of the functional decline of ISCs and reduces the overall fitness of *Drosophila* upon aging.

In addition, we have also analyzed the lifespan of *Drosophila* carrying *esg^ts^>UAS cad RNAi* (flies were cultured at 18°C for 25 days, then transferred to growth at 29°C to induce cad RNAi). However, we didn't find a significant increase (but a minor reduction) in the longevity of flies with *cad* depletion compared to the control flies *(esg^ts^>UAS-GFP)* (S7G Fig). The possible reason that knockdown *cad* does not increase fly lifespan may be because that depletion of *cad* causes over-reduction of ISCs under normal conditions or *cad* has other unknown functions in ISCs/progenitor cells.

## Discussion

Over the past decades, studies on the homeobox gene *cad* mainly focused on exploring its roles in antero-posterior (A-P) patterning, organ morphogenesis during development, tumorigenesis, and the innate immune response [38,39,41,57–61]. It has been reported that the transcriptional expression of *cad* increases in the midguts of both young *Drosophila* after injury and aged *Drosophila* [43,44]. However, the function of *cad* in the context of ISCs and aging remains largely unexplored. This study showed that the expressions of *cad* in ISCs and EBs of aged *Drosophila* midguts were significantly upregulated compared with the expressions in ISCs and EBs of young *Drosophila* midguts. Importantly, the aging-related CAD increase in ISCs and EBs contributes to gut hyperplasia in the midguts of aged *Drosophila* (Fig 7R).

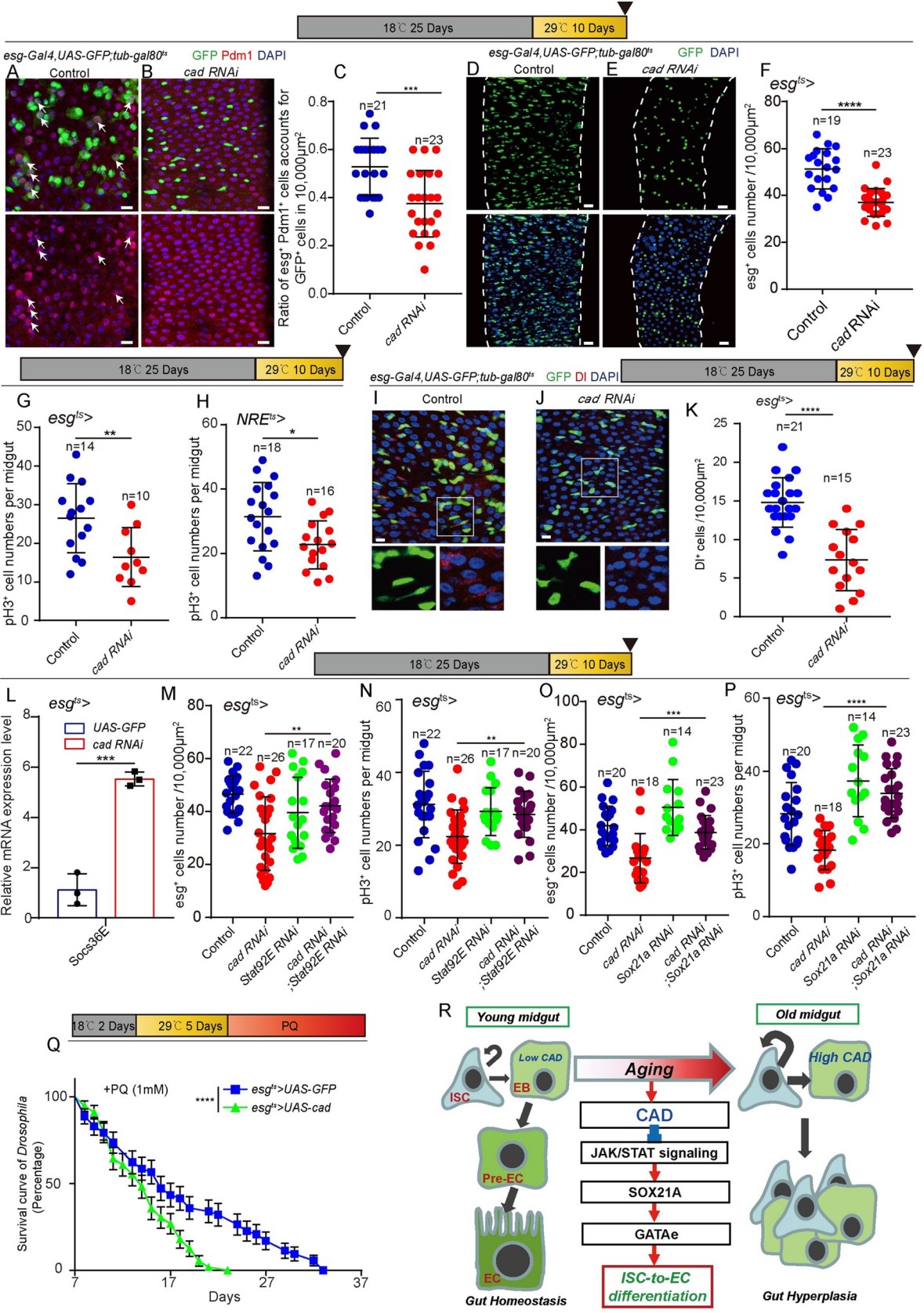

**Fig 7. Reduction of *cad* expression in ISCs and progenitor cells represses age-associated gut hyperplasia in *Drosophila*.** (A-B) Immunofluorescence images of *esg*-GFP (green) and Pdm1 (red) staining with midgut from the PMG in 35-day-old (25 days at 18˚C, then 10 days at 29˚C) control *Drosophila* (A, *esg^ts*-Gal4>UAS-GFP) and 35-day-old *Drosophila* carrying *esg^ts*-Gal4>*cad* RNAi (B). *esg*-GFP (green) represents ISCs and their differentiating cells. Pdm1 staining (red) was used to visualize differentiating ECs. White arrows indicate differentiating pre-ECs (*esg*-GFP⁺ and Pdm1⁺ cells). *esg*-GFP⁺ and Pdm1⁻ cells are ISCs or EBs. *esg*-GFP⁻ and Pdm1⁺ cells are mature ECs. (C) Quantification of the ratio of *esg*-GFP⁺ Pdm1⁺ cells per 10,000 μm² area from PMG of control *Drosophila* with genotypes as indicated in (A-B). The number n is indicated. Each dot represents one midgut. Each dot corresponds to one ROI (10,000 μm² area). (D-E) Immunofluorescence images of midgut sections from the R4 region in 35-day-old (25 days at 18˚C, then 10 days at 29˚C) control *Drosophila* (D, *esg^ts*-Gal4>UAS-GFP) and *Drosophila* carrying *esg^ts*-Gal4>*cad* RNAi (E). *esg*-GFP (green) indicates ISCs and their differentiating cells. (F) Quantification of *esg*-GFP⁺ cell numbers in experiments (D-E). The number n represents the ROI in midguts from each experiment. One dot corresponds to one ROI (10,000 μm² area). (G-H) Quantification of pH3⁺ cell numbers in midguts from 35-day-old (25 days at 18˚C, then 10 days at 29˚C) control *Drosophila* (*esg^ts*-Gal4>UAS-GFP in G and *NRE^ts*-Gal4>UAS-GFP in H), *Drosophila* carrying *esg^ts*-Gal4>*cad* RNAi (G), and *NRE^ts*-Gal4>*cad* RNAi (H). The number n represents the whole midguts. One dot corresponds to one midgut. (I-J) Immunofluorescence images of midgut sections from the R4 region in 35-day-old (25 days at 18˚C, then 10 days at 29˚C) control *Drosophila* (I, *esg^ts*-Gal4>UAS-GFP) and *Drosophila* carrying *esg^ts*-Gal4>*cad* RNAi (J). *esg*-GFP (green) indicates ISCs and their differentiating cells. Dl (red) staining was used to visualize ISCs. (K) Quantification of Dl⁺ cell numbers in experiments (I-J). The number n represents the ROI in midguts from each experiment. One dot corresponds to one ROI (10,000 μm² area). (L) Relative mRNA fold changes of *Socs36E* in sorted *esg*⁺ cells from midguts of aged (25 days at 18˚C add 10 days at 29˚C) control *Drosophila* (blue lines, *esg^ts*-Gal4>UAS-GFP) and aged *Drosophila* with cad depleted (red lines, *esg^ts*-Gal4>*cad* RNAi). The changes of expressions were plotted relative to the control *Drosophila*, which was set to 1. (M-N) Quantification of *esg*⁺ (M) or pH3⁺ (N) cell numbers in midguts from 35-day-old (25 days at 18˚C, then 10 days at 29˚C) control *Drosophila* (*esg^ts*-Gal4>UAS-GFP), *Drosophila* carrying *esg^ts*-Gal4> *cad* RNAi, and *Drosophila* carrying *esg^ts*-Gal4> *cad* RNAi; *Stat92E* RNAi. The number n represents the whole midguts (N). One dot corresponds to one midgut in N. The number n represents the ROI in midguts from each experiment in M. One dot corresponds to one ROI (10,000 μm² area) in M. (O-P) Quantification of *esg*⁺ (O) or pH3⁺ (P) cell numbers in midguts from 35-day-old (25 days at 18˚C, then 10 days at 29˚C) control *Drosophila* (*esg^ts*-Gal4>UAS-GFP), *Drosophila* carrying *esg^ts*-Gal4> *cad* RNAi, and *Drosophila* carrying *esg^ts*-Gal4-driven *cad* RNAi and *Sox21a* RNAi. The number n represents the whole midguts in P. One dot corresponds to one midgut in P. The number n represents the ROI in midguts from each experiment in O. One dot corresponds to one ROI (10,000 μm² area) in O. (Q) Survival rate of control *Drosophila* (*esg^ts*-Gal4-driven UAS-GFP) and *Drosophila* carrying *esg^ts*-Gal4> UAS-cad under PQ treatment (see methods). (R) Schematic model of the underlying mechanism. In young *Drosophila*, CAD expresses at a basal level in ISCs and EBs. The basal expressed *cad* prevents the overexpression of SOX21A and GATAe transcription factors in ISCs and EBs, which prevents the ISCs from producing ECs unless the intestinal epithelia are damaged. CAD represses SOX21A and GATAe expression in ISCs and EBs by inhibiting the activation of JAK/STAT signaling. The expression of CAD increases in ISCs and EBs of *Drosophila* during aging. Ultimately, this leads to a decrease of differentiation efficiency of ISCs and EBs with age, which, in turn, results in gut hyperplasia as indicated by the continuous accumulation of ISCs and their differentiating progenies (i.e., *esg*⁺ polyploid cells) in the midguts of aged *Drosophila*. DAPI stained nuclei are shown in blue. Scale bars represent 10 μm (Fig 7A, 7B, 7I and 7J) or 20 μm (Fig 7D and 7E). Error bars represent SD. Student's t-tests were used to assess statistical significance. For the survival test, the log-rank test was used to analyze the statistical significance: *p < 0.05, **p < 0.01, ***p < 0.001, ****p < 0.0001, and NS (non-significant), which represents p > 0.05. See also S7 Fig.

In addition to the regulation of the formation of body sections during *Drosophila* embryo-genesis [39,57,58], *cad* has also been shown to repress the expressions of antimicrobial peptides (AMPs) in *Drosophila* midguts [41] and to function in the regulation of innate immune homeostasis [41,62–66]. Although the immune function of *cad* in intestinal epithelia has been well studied [41,62–66], the roles of *cad* in ISCs and progenitor cells and how it regulates gut epithelial homeostasis were largely ignored. This study showed that deletion of *cad* in EBs induced EB-to-EC differentiation, while overexpression of *cad* inhibited EC formation. Using the *Drosophila* genetic strategy, this study demonstrated that CAD functions in EBs to regulate ISC-to-EC differentiation by modulating the JAK/STAT-SOX21A-GATAe cascade (Fig 7R).

In this study, we showed that the expression of *cad* was significantly upregulated in ISCs and EBs of aged *Drosophila*. The failure of ISC-to-EC differentiation, and thus inducing gut hyperpla-sia during aging may attribute to excessive *cad* expression. But how the expression of *cad* in aged animals is regulated remains unclear. *Cad* functions as an effector of induction of AMPs in immune homeostasis have been reported[61,66]. Besides its roles in the immune system, oxida-tive stress has also been shown to regulate the expression of *cad* [43,67]. It is conceivable that these dynamic changes of *cad* expression during aging could be influenced by these external phys-iological factors since the levels of reactive oxygen species and oxidative damage increase with age, and the innate immune response tends to become hyperactive in older organisms [68,69]. On the other hand, the *cad*-related homeobox gene has been reported to be regulated by Relish and NF-κB pathway, while the activation of NF-κB has been firmly linked to the age-related

inflammatory processes under increased oxidative stress [43,70–72]. DRE/DREF is another molecular effector that has been reported to regulate the transcription of *cad* [73]. Previous studies have indicated that DREF activity is modulated by the intracellular redox state [74]. These intriguing correlations imply to us that the age-associated changes of expression of *cad* may probably be the result of alters of these upstream molecular effectors and related signaling pathways caused by the accumulation of immune damage and oxidative stress during aging. Understanding the dynamic changes of *cad* expression under young and aging conditions may be useful for us to clarify the mechanisms through which external stresses result in the differentiation disorder of ISC in the aging process, and then lead to intestinal function degeneration and tumorigenesis.

In this study, we showed that high levels of *cad* expression in ISCs and EBs results in the failure of ISC-to-EC differentiation during aging. This defect leads to the continuous accumulation of differentiation defective EBs and immature ECs, and consequently tumorigenesis. There are some cytokines/peptides, like Hh, Upd1, Upd2 produced by ISCs/EBs to maintain local tissue homeostasis [75–77]. Elevated Hh and JAK-STAT signaling pathways are essential for the regenerative proliferation of ISCs. However, excessive secretion and ectopic activation of Hh signals may be a stimulator in many types of cancers, including lung, prostate, pancreas, liver, bladder, and skin [78–83]. Moreover, the over-release of insulin-like factor Ilp6 and TNF ligand Eiger derived from stem/progenitors may also induce colorectal cancer, diabetic enteropathy, and commonly triggers chronic systemic inflammation [84–86]. Aside from the uncontrolled retention of transient undifferentiated cells, another consequence of ISC-to-EC differentiation defect is the ablation of mature ECs in the midguts. As one of the differentiated cells in *Drosophila* intestine, ECs are responsible for the absorption of nutrients, and the secretion of digestive enzymes. Unable to produce mature ECs may directly lead to the interruption of normal digestion, absorption, and transport of a variety of nutrients, which results in malnutrition, diarrhea, steatorrhea, and weight loss, named malabsorption syndrome clinically [87,88]. Poor intestinal absorption and secretion also have been associated with a variety of different human diseases, such as coeliac disease, lactase deficiency, or Whipple's disease [88]. In addtion, lack of ECs also contributes to the disruption of the gut barrier, which increases intestinal permeability. This dysfunctional intestinal barrier in turn facilitates the translocation of harmful substances and pathogens to the bloodstream, and thus impacts on other distal organs and gives rises to pathology and physiology of several systemic disorders, such as inflammation, allergic diseases, chronic kidney disease, hyperglycemia, and Alzheimer's disease [89–94].

In aged *Drosophila*, midguts showed dysplasia, caused by both ISC hyper-proliferation [19–21] and aberrant differentiation [20,28,55]. Several signaling pathways, such as JNK [20], insulin receptor [27–30], p38-MAPK [19,22], ROS [23], and unfolded protein response (UPR) [25,26], have been identified to regulate aging-associated ISC hyper-proliferation. However, the causes of the aberrant differentiation of ISCs and progenitor cells upon aging remain largely unexplored. This study demonstrated that the upregulation of the homeobox gene *cad* resulted in a decline of ISC-to-EC differentiation in *Drosophila* midguts upon aging. These findings identify a novel regulator that specifically modulates ISC-to-EC differentiation during gut epithelial regeneration and aging.

## Materials and methods

### *Drosophila* breeding and maintenance

The stocks of *Drosophila* were fed on standard yeast food (cornmeal 50 g, yeast 18.75 g, sucrose 80 g, glucose 20 g, agar 5 g, and propionic acid 30 ml combined in 1 L water) at 25˚C, 65% humidity, and on a 12-h light/dark daylight cycle unless otherwise stated, unless specifically mentioned.

The temporal and regional gene expression targeting (TARGET) method [95] was used to manipulate the *Gal4-UAS*-mediated RNAi and overexpression experiments in this study. To repress the *GAL4* system, the crosses were maintained at 18˚C when driving temperature-sensitive *GAL4*-mediated RNAi or gene overexpression. When flies eclosion or at a certain age after flies eclosion, the adults were shifted to 29˚C to turn on the *GAL4* system, which induces RNAi or gene overexpression. Eclosed flies were incubated at 29˚C for indicated days followed by dissection of midguts for immunostaining or western blot. In this study we used the mated females for experiments on the *Drosophila* midguts.

For the aging-associated experiments, the *tub-Gal80*[ts] transgenic *Drosophila* combined with *esg-Gal4* or *NRE-Gal4* were grown at a permissive temperature (18˚C) to limit Gal4 activity. Adult flies continue to raise in 18˚C for 25 days after eclosion then transferred to a non-permissive temperature (29˚C) for 10 days until dissection midguts for immunostaining or western blot analyses.

## *Drosophila* lines in this study

The following *Drosophila* lines were obtained from the TsingHua Fly Center (THU, Beijing, China): *FRT40A* (Thu0389), *10×Stat-GFP* (Thu0274), and *GATAe* RNAi (Thu1490). The following *Drosophila* line was obtained from the Zurich ORFeome Project (FlyORF): *UAS-caudal-3×HA* (F000471). The following *Drosophila* lines were obtained from the Bloomington *Drosophila* stock center (BDSC): *cad* RNAi (BDSC# 57546 and BDSC# 34702), *w*[1118] (BDSC# 3605), *UAS-Sox21a* (BDSC# 68155), *cad-EGFP* (BDSC# 30875), *Notch* RNAi (BDSC# 7078), and *cad*[2] *FRT40A* (BDSC# 7091). The following *Drosophila* line was obtained from the Vienna *Drosophila* Resource Center (VDRC): *Stat92E* RNAi (v106980), *rpr* RNAi (v101234).

The following *Drosophila* lines were kindly provided as indicated in the following: *tub-GAL line*, *esg-GFP*, *esg-Gal4* and *UAS-lacZ* line, by Dr. Allan Spradling; *esg*[ts]*-Gal4* line [5], *MyoIA*[ts]*-Gal4* line [18], *ISC*[ts]*-Gal4* line [51], *NRE*[ts]*-Gal4* line [51], and *UAS-hop*[TUM] line by Dr. Benjamin Ohlstein [52]; *esg-lacZ* line by Dr. Mark Van Doren [96]. The following *Drosophila* lines were available in our laboratory: *Sox21a-HA* lines [52] and *UAS-GATAe-HA* line [52]. All *Drosophila* lines used in this study are listed in S3 Table.

The transgenic *Drosophila* line *UAS-cad-HA* was constructed in our laboratory. In brief, cDNA of *cad* sub-cloned was used into the pEntry-3xHA vector [52] through the pEASY-Uni seamless cloning and assembly kit (TransGen Biotech, CU101-02). Then, the previous product was sub-cloned into the pTW vector (obtained from Allan Spradling Lab) through LR recombination reaction. Finally, the vectors were cloned and tested through sequencing, and sent to UniHuaii Corporation (Zhuhai, China) for *Drosophila* egg injection. The *cad* cDNA used the following primers: *Cad L*: 5' ATGGTTTCGCACTACTACAA 3'; *Cad R*: 5' TCACATCGAGAGCGTGCCCA 3'.

To obtain the *Drosophila* line *UAS-cad*. The transgenic *Drosophila* line *UAS-cad* was constructed in our laboratory. In brief, cDNA of *cad* sub-cloned was used into the pEntry vector [52] through the pEASY-Uni seamless cloning and assembly kit (TransGen Biotech, CU101-02). Then, the previous product was sub-cloned into the pTW vector (obtained from Allan Spradling Lab) through LR recombination reaction. Finally, the vectors were cloned and tested through sequencing, and sent to UniHuaii Corporation (Zhuhai, China) for *Drosophila* egg injection. The *cad* cDNA used the following primers: *Cad L*: 5' ATGGTTTCGCACTACTA-CAA 3'; *Cad R*: 5' TCACATCGAGAGCGTGCCCA 3'.

## Mosaic analysis with repressible cell marker clone induction

To generate *cad*[2] [97] mutant MARCM clones, the *cad*[2] mutant carrying the *FRT40A* [98]line was obtained from the BDSC (BDSC# 7091). To generate *UAS-cad* MARCM clones, *FRT40A* was crossed to *UAS-cad* to produce the following *Drosophila*: *FRT40A/cyo*; *UAS-cad /TM6B*.

Then, these *Drosophila* were crossed to *yw hsFLP::tub-Gal4::UAS-nls GFP/FM7; tubG80 FRT40A/CyO* (gifted from Allan Spradling) was used to obtain *yw hsFLP::tub-Gal4::UAS-nls GFP/+; tubG80 FRT40A/ FRT40A::cad²*, and *yw hsFLP::tub-Gal4::UAS-nls GFP/+; tubG80 FRT40A/ FRT40A; UAS-cad/+* flies. The crosses were kept at 25˚C. Three-day-old adult flies were heat-shocked for 1 h twice at 37˚C to induce clones and then kept at 29˚C for 7 days until dissection. *UAS-cad* was only expressed in the clone which presented GFP.

For MARCM clone production, adult *Drosophila* were fed at 25˚C for 2–3 days and heat-shocked at 37˚C for 1 h twice. After heat-shock treatment, *Drosophila* were kept at 25˚C, and then, these flies were dissected and observed at 7 days ACI [52].

## Paraquat and DSS treatment

For the Paraquat (PQ) or DSS treatment, *Drosophila* were starved in empty vials for 2 h and were exposed in 5% (wt/vol) sucrose with 10 mM PQ (Aladdin, China) or 5% (wt/vol) DSS (MP Biomedicals, CAS: 9011-18-1, 36~50KD) for 24 h, followed by recovery for 24 h. Then, *Drosophila* were dissected in PBS. *Drosophila* were transferred from medium to empty bottles for 2 h. Filter paper was cut into $3.5 \times 6.0$ cm pieces, and infiltrated in 5% (wt/vol) sucrose with 10 mM PQ or with 5% (wt/vol) DSS. Then, the moist paper was added to the empty bottles for 24 h, and *Drosophila* were transferred into new standard medium without PQ or DSS. Identical *Drosophila* fed with 5% sucrose were used as control [52]. The PQ- or DSS-induced damages experiments were performed in the temperature incubator at 25˚C, unless specifically mentioned.

For manipulation of the PQ-treatment RNAi or overexpression experiments, the GAL4ts lines and UAS lines were maintained at 18˚C to repress the GAL4 system. One day after eclosion, the adults were shifted to 29˚C to activate the GAL4 system, which induces RNAi or overexpression. Eclosed flies were incubated at 29˚C for 5 days. Then *Drosophila* were starved in empty vials for 2 h and were exposed in 5% (wt/vol) sucrose with 10 mM PQ for 24 h, followed by recovery for 24 h. The controls were treated in the same way in parallel. The process of PQ treatment was manipulated at 29˚C.

## Immunostaining and microscopy

Guts were dissected in PBS, and were immersed in PBS with 4% EM-paraformaldehyde (PBS formula: 100 mM glutamic acid, 26 mM KCl, 20 mM MgSO$_4$, 4.5 mM Na$_2$HPO$_4$, 1 mM MgCl$_2$, pH 7.4) at room temperature (25˚C) for 45 min. Guts were washed in PBST (PBS, 0.1% Triton X-100) 3 times, 15 min each, followed by incubation with primary and second antibodies in PBST buffer after soaking in BSA buffer (PBS, 0.5% BSA, 0.1%Triton X-100) for 30 mins.

The following primary antibodies were used: Chicken polyclonal anti-GFP (Abcam, AB_300798, 1:1000), rabbit polyclonal anti-GFP (Proteintech, AB_11042881, 1:1000), rabbit anti-HA (Cell Signaling Technology, AB_1549585, 1:1000), mouse anti-Delta (anti-Dl) (Developmental Studies Hybridoma Bank, AB_528194, 1:100), rabbit anti-Pdm1 (from Dr. Xiao-Hang Yang, 1:500), and mouse anti-Prospero (Developmental Studies Hybridoma Bank, RRID: AB_528440). The method of gut immunostaining with Dl used methanol-heptane and methanol to replace the PFA and PBST before BSA incubation. The secondary antibodies (Alexa 488, Alexa 568, and Alexa 647, purchased from MolecularProbes/Invitrogen) were diluted and used at 1:2000. The final concentration of 4,6-diamidino-2-phenylindole (DAPI; Sigma) was 1 μg/ml.

Confocal images were acquired by a Leica TCS-SP8 confocal microscope. Images for each set of experiments were acquired as confocal stacks under the same settings. These settings

including the distinguish at 1024x1024 and using bi-direction scanning. The gain value up reaches 800 and off-set arrange from -0.1%~-0.2% in each image. To illuminate the Z-axis detail staining and get clear Images, they were acquired as a confocal z-stacks number usually range from 5 to 10 and finally projected to one image. Adobe Photoshop and Adobe Illustrator CS5 were used to assemble the images. The number of midgut cells in all quantifications was counted using a Leica DM6-B microscope.

## Guts lysate and Western blotting analyses

Dissected *Drosophila* midguts, mixed in RIPA Lysis Buffer (P0013B, Beyotime Biotechnology) with appropriate protease inhibitors, then transferred to liquid nitrogen. Put the tissues on ice 30 minutes after used grinding rod for homogenization. Collected lysate supernatant transferred to total proteins through BCA kit (PierceRapid Gold BCA Protein Assay Kit, Thermo, A53226). Each sample added 5× loading buffer to boiled 12 min. Transferred the protein gel to polyvinylidene difluoride (PVDF) membranes and used 5% TBST with defatted milk to blocked 1 h. After blocking, incubated in primary antibody for whole night at 4˚C and used TBST to wash 3 times at room temperature 25˚C, each time 10 min. Shaking PVDF membranes with horseradish peroxidase-labelled secondary antibody at room temperature 25˚C for incubation.

   The Western blotting antibodies were shown in the Table of Antibodies List. The secondary antibodies were horseradish peroxidase-conjugated Goat anti-mouse (Beyotime Biotechnology, #A0216, 1:1000), Goat anti-rabbit (Thermo, #A0208, 1:1000).

## FACS and qRT-PCR

200 midguts were dissected from young or old female flies, respectively. Flies were dissected in 4˚C diethyl pyrocarbonate (DEPC)-treated water-PBS, then incubated in 1 mg/ml Elastase (Sigma, cat. no. E0258) combined DEPC-PBS mixture at room temperature, mixed the sample softly 5 times every 15 min during incubation. After dissociation, the samples were centrifuged at 400×g for 20 min at 4˚C, then re-suspended in 4˚Cdiethyl pyrocarbonate (DEPC)-treated water-PBS, filtered with 70 mm filters (Biologix) and sorted by FACS Aria II sorter (BD Biosciences). GFP$^+$ cells in the *esg-GFP Drosophila* midguts were sorted out, the midguts of $w^{1118}$ flies were set as fluorescence gate. We used three biological replicates to detect, each replicate contains 50,000 esg-GFP+ cells, the total RNA was analyzed in the Arcturus PicoPure RNA isolation kit (Applied Biosystems) follow the protocol. We used the PrimeScript RT reagent Kit (TaKaRa) to synthesize cDNA, RNA was reverse-transcript by oligo dT, the first-strand cDNA was diluted with sterile water 50 times and transfer to real-time PCR. The Real-time PCR was accomplished in Quant-Studio 5 System (Thermo Fisher Scientific), each sample employs SYBRGreen (Genestar). The reference standard group is Rp49, method of expression value calculation is $2^{-\triangle\triangle CT}$. The expression of the standard sample was normalized to 1, All of the primer sequences of qPCR are available upon reasonable request.

## RNA-Seq and data analysis

The *Drosophila* crosses were cultured on normal food at 18˚C. *esg$^{ts}$>UAS-lacZ Drosophila* were used as the control for the *cad*-depleted *Drosophila* (*esg$^{ts}$>cad RNAi*). To induce RNAi, *Drosophila* were maintained at 29˚C for 7 days. About 100 female flies for each biological replicate were dissected in PBS on ice. Midguts were frozen on drikold and isothiocyanate-alcohol-phenyl-chloroform was used to collect total RNA after dissection. Then, the total RNA was sent to Berry Genomics Corporation (China) for sequencing on a novaseq 6000 platform (Illumina, San Diego, CA, USA), and quality control was performed by FastQC (version 0.11.8).

The raw data of RNA-seq read lengths of 150 bp were aligned to the *Drosophila* reference genome (Ensembl BDGP6 release-89). The aligned reads (sam files) were transferred to bam files and sorted by SAMtools. Gene expressions in different samples were determined by DESeq2 (version 1.22.2). The differentiation of expression was verified if $p \leq 0.05$ following a Benjamini and Hochberg correction for multiple hypothesis testing.

## Fluorescence intensity statistics

Immunofluorescence images were analyzed via Confocal microscopy. The fluorescence intensity statistics in the region of interest (ROI) was calculated using ImageJ. The detailed process of the fluorescence intensity statistics was previously described [52]. The steps in brief:

Open image: File -> Open.

Split Channels: Image-> Color -> Split Channels. Select the channel which needs to calculate and strike out redundant channels.

Scale Setting: Analyze-> Set scale-> click to remove scale. Set scale to determine the unit of length is Pixel.

Measurements Setting: Analyze-> Set Measurements. Choose the boxes named "Area", "Integrated Density", and "limit to threshold". Click on the "OK" box after chosen.

Choose the ROI or Cell by different kinds of drawing/selection tools and select another smaller region around the ROI or cell as background.

Choose different types of ROI, Cell and background regions through using boxes named"ROI Manager".

Select the boxes named"Measure", calculate the integrated density.

Integrated Density = Integrated Density of ROI or cell—Integrated Density of background region/Area of background region × Area of ROI or cell.

## Band Gray Value analysis of Western blotting

Band Gray Value analysis of Western Blotting was used the software ImageJ. The specific methods are shown:

File -> Open. Choose the selected image.

Edit -> Invert.

Analyze -> Set Scale -> Choose the image scale to remove. Set scale to determine the unit of length is Pixel.

Analyze -> Set Measurements. Choose the bottom of "Area" ->"Mean Gray Value", then limited the number to attain threshold, Choose the botton of "OK".

Pick up the region of band and choose a smaller area as the background.

Choose the ROI Manager to select the different regions of band and background region.

Click the box of "Measure" to compute the average value of Gray Value.

(Band Gray Value = (Mean Gray Value of band region × Area of Band Region)-(Background Mean Gray Value × Area of background Region))

## Lifespan assays under normal and stressful conditions

For survival tests under normal conditions, 100 female flies (2 days old) with the same phenotype were collected and fed on the temperature incubator at 18˚C to repress the GAL4 system. Two days after fly eclosion, the adults were transferred to 29˚C temperature turn on GAL4 activity for survival tests analysis. Living flies were counted and transported to new food tubes every 2 days. The survival tests were repeated 3 times as 3 independent experiments.

For survival tests under chronic oxidative damage conditions, 100 female flies (2 days old) with the same phenotype were collected and fed on the temperature incubator at 18˚C to

repress the GAL4 system. Two days after fly eclosion, the adults were transferred to 29˚C temperature turn on GAL4 activity for 5 days. In order to finish the viability test, 100 female flies (2 days old) have same genetic background were collected and divided into five tubes with food medium. Transferred flies into empty tubes for starvation at 29˚C before Viability Tests. Then put the filter paper (3.7×5.8 cm) saturated with 1 mM PQ in 5% sucrose solution at 29˚C. The number of flies alive still counted, the filter paper saturated with 1 mM PQ in 5% sucrose solution exchanged continually every day.

## Quantification of statistical analysis

GraphPad Prism v7.0 was used to evaluate the statistical significance after verifying normality and equivalence of variances. Data in this study are presented as the means ± standard deviation (SD) from at least three independent biological replicates unless otherwise specified. Statistical significance was determined using two-tailed Student's t-tests unless otherwise specified. Significance is stated in the text or figure legends. For all tests, a $p < 0.05$ was considered statistically significant.

## Software availability

The software package R (version 3.5.3, download from https://www.r-project.org/) was used for downstream analysis of RNA-seq results. The custom ImageJ used for immunofluorescent staining and Western blotting quantification (download from https:https://imagej.nih.gov/ij/). Prism 7.0 (GraphPad) was also used in this study (download from https://www.graphpad.com/).

## Supporting information

**S1 Fig. *Cad* highly expresses in the posterior midgut and can be induced by injury.** (A) Cartoon model of *Drosophila* ISC lineages. An ISC (Dl[+] and *esg*[+]) undergoes asymmetric division once to produce a new ISC and a diploid precursor enteroblast (EB; *esg*[+] and *NRE*[+]) or a diploid precursor enteroendocrine mother cell (EMC; *esg*[+] and Pros[+]). The post-mitotic EB further differentiates into premature enterocytes (pre-ECs) (*esg*[+] and Pdm1[+]), which continue to differentiate into octoploid mature ECs (Pdm1[+]). The EMC divides once to produce a pair of diploid enteroendocrine cells (EEs; Pros[+]). (B-C) The *cad* gene reads per kilobase per million mapped reads (RPKM) values in ISCs (B) and EBs (C) of five regions from the midguts of *Drosophila*. Expression was determined using RNAseq in experiments described in [44]. (D) The *cad* gene RPKM values in ISC, EB, EC, and EE from R1-R5 of midguts. Expression was determined using RNAseq in experiments described in [44]. (E) Relative mRNA fold change of *cad* in sorted *esg*-GFP[+] cells of young (green bar) and aged (red bar) *Drosophila* (*esg*-GFP/CyO). The *cad* expressions in *esg*-GFP[+] cells of *Drosophila* with different ages are plotted relative to 5-day *Drosophila*, which was set to 1. Error bars indicate the standard deviation (SD) of three independent experiments. (F) Western blot result of CAD expression in midguts from young (age 5 days) and old (age 30 days). (G) The whole posterior midgut immunofluorescence images of CAD-EGFP (green) staining from young (age 7 days) and old (age 30 days) *Drosophila* carrying CAD-EGFP (green). (H-I) Immunofluorescence images of CAD-EGFP (green) staining with the midgut section from R1-R5 region (except R4 which were showed in Fig 1) of young (7 days; H) and old (30 days; I) *Drosophila* carrying CAD-EGFP midgut. (J) Immunofluorescence images of CAD-EGFP (green) staining with the midgut section from the PMG (the posterior midgut) of young *cad-EGFP Drosophila* which were treated with 5% sucrose (used as control), PQ, or DSS, respectively. (K) Quantifications of fluorescence intensity of CAD-EGFP in per region of interest (ROI) from the PMG as shown in (J). The number n is

indicated. Each dot represents one ROI (10,000 μm$^2$). DAPI stained nuclei are shown in blue. Scale bars represent 50 μm (S1G Fig), 25 μm (S1H, S1I, and S1J Fig), 10μm (S1J Fig). Error bars represent SD. Student's t-tests were used to assess significance: $^*$p < 0.05, $^{**}$p < 0.01, $^{***}$p < 0.001, $^{****}$p < 0.0001, and NS (non-significant), which represents p > 0.05. (TIF)

**S2 Fig. The effects of *cad* in the regulation of ISC proliferation, differentiation, and survival.** (A) Relative mRNA fold change of *cad*-depleted *Drosophila* by *tub-Gal4*-driven two different *cad* RNAi lines. The *cad* expressions in *Drosophila* with different RNAi are plotted relative to control flies (*tub-Gal4*), which was set to 1. Error bars indicate the standard deviation (SD) of three independent experiments. (B-C) Immunofluorescence images of *esg*-GFP (green) and Pdm1 (red) staining with the midgut under normal conditions. The midgut section from AMG of control flies (B, *esg$^{ts}$-Gal4*-driven *UAS-GFP*) and AMG of *Drosophila* carrying *esg$^{ts}$-Gal4*-driven *cad RNAi* (C). *esg*-GFP$^+$ and Pdm1$^-$ cells are ISCs or EBs. *esg*-GFP$^-$ and Pdm1$^+$ cells are matured ECs. (D) Quantification of the ratio of *esg*-GFP$^+$ and Pdm1$^+$ cells per 10,000 μm$^2$ area of the AMG as indicated in (B-C). The number n represents counted ROI in midguts from each experiment. Each dot corresponds to one ROI (10,000 μm$^2$ area). (E-H) Quantification of the *esg$^+$* cells (E-F) and the *Dl$^+$* cells (G-H) per 10,000 μm$^2$ area of the anterior midgut (AMG; E and G) and the posterior midgut (PMG; F and H) from control *Drosophila* (*esg$^{ts}$-Gal4*-driven *UAS-GFP*), *Drosophila* carrying *esg$^{ts}$-Gal4>cad* RNAi, and *Drosophila* carrying *esg$^{ts}$-Gal4>UAS-cad-HA*, under homeostatic conditions. The number n is indicated. Each dot corresponds to one ROI (10,000 μm$^2$ area). (I-L) Quantification of the *esg$^+$* cells (I-J) and the *Dl$^+$* cells (K-L) per 10,000 μm$^2$ area of the AMGs (I and K) and the PMGs (J and L) from control *Drosophila* (*esg$^{ts}$-Gal4*-driven *UAS-GFP*), *Drosophila* carrying *esg$^{ts}$-Gal4>cad* RNAi, and *Drosophila* carrying *esg$^{ts}$-Gal4>UAS-cad-HA*, under PQ-treated conditions (PQ-REC-1D). The number n is indicated. Each dot corresponds to one ROI (10,000 μm$^2$ area). (M-N) Immunofluorescence images of *esg*-GFP (green) and TUNEL (red) staining with the midgut under normal conditions. The midgut section from PMG of control flies (M, *esg$^{ts}$-Gal4*-driven *UAS-GFP*) and *Drosophila* carrying *esg$^{ts}$-Gal4*-driven *cad RNAi* (N). *esg*-GFP (green) identifies ISCs and their differentiating cells. White arrowhead indicates cells in apoptosis. (O-Q) Immunofluorescence images of *esg*-GFP (green) and Dl (red) staining with the midgut section from the R4 region of *Drosophila* carrying *esg$^{ts}$-Gal4>UAS-GFP* (O), *Drosophila* carrying *esg$^{ts}$-Gal4> cad RNAi* (P), and *Drosophila* carrying *esg$^{ts}$-Gal4>cad RNAi* and *rpr RNAi* (Q). *esg*-GFP (green) represents ISCs and their differentiating cells. Dl staining (red) was used to visualize ISCs. White arrows indicate Dl$^+$ ISCs. (R) Quantification of the Dl$^+$ cell numbers per 10,000 μm$^2$ area of midgut as indicated in (O-Q). The number n represents counted ROIs in midguts from each experiment. Each dot corresponds to one ROI (10,000 μm$^2$ area). DAPI stained nuclei are shown in blue. Scale bars represent 10 μm (S2B, S2C, S2M, S2N and S2O–S2Q Fig). Error bars represent SD. Student's t-tests were used to assess significance: $^*$p < 0.05, $^{**}$p < 0.01, $^{***}$p < 0.001, $^{****}$p < 0.0001, and NS (non-significant), which represents p > 0.05. (TIF)

**S3 Fig. *Cad* functions in EBs to prevent ISCs to produce differentiated ECs.** (A-B) Immunofluorescence images of *esg*-GFP (green) and Pdm1(red) staining with the midgut section from the R4 region of control *Drosophila* (A, *esg$^{ts}$-Gal4>UAS-GFP*) and *Drosophila* carrying *esg$^{ts}$-Gal4>UAS-cad* (B), under normal conditions. *esg*-GFP (green) represents ISCs and their differentiating cells. Pdm1 staining (red) was used to visualize matured ECs. (C) Quantification of the ratio of *esg*-GFP$^+$ and Pdm1$^+$ cells per 10,000 μm$^2$ area of the midgut as indicated in (A-B). The number n represents counted ROIs in midguts from each experiment. Each dot

corresponds to one ROI (10,000 μm$^2$ area). (D-E) Immunofluorescence images of *NRE*-GFP (green) and Pdm1(red) staining with the midgut section from the R4 region of control *Drosophila* (D, *NRE$^{ts}$-Gal4>UAS-GFP*) and *Drosophila* carrying *NRE$^{ts}$-Gal4>UAS-cad* (B), under normal conditions. *NRE*-GFP (green) represents EBs. Pdm1 staining (red) was used to visualize matured ECs. (F) Quantification of the ratio of *NRE*-GFP$^+$ and Pdm1$^+$ cells per 10,000 μm$^2$ area of the midgut as indicated in (D-E). The number n represents counted ROIs in midguts from each experiment. Each dot corresponds to one ROI (10,000 μm$^2$ area). (G-H) Immunofluorescence images of *ISC*-GFP (*ISC$^{ts}$-Gal4*-driven *UAS-GFP*; green) and Pdm1 (red) staining with the midgut section from the R4 region of control flies (G, *ISC$^{ts}$-Gal4*-driven *UAS-GFP*) and *cad*-depleted *Drosophila* by *ISC$^{ts}$-Gal4*-driven *cad* RNAi (H). *ISC*-GFP (green) indicates ISCs. Pdm1 staining (red) was used to visualize differentiating ECs. *ISC*-GFP$^+$ and Pdm1$^-$ cells are ISCs. *ISC*-GFP$^-$ and Pdm1$^+$ cells are mature ECs. (I) Quantification of the ratio of *ISC*-GFP$^+$ and Pdm1$^+$ cells per 10,000 μm$^2$ area of the R4 region of midguts as shown in (G-H). The number n represents counted regions of interest in midguts from each experiment. Each dot corresponds to one region of interest (ROI = 10,000 μm$^2$ area). (J-K) Immunofluorescence images of *ISC*-GFP (green) and Pdm1 (red) staining with the midgut treated with PQ-REC-1D. The midgut section from the R4 region of control flies (J, *ISC$^{ts}$-Gal4*-driven *UAS-GFP*) and *cad*-overexpressing *Drosophila* by *ISC$^{ts}$-Gal4*-driven *UAS-cad-HA* (K). *ISC*-GFP (green) represents ISCs. Pdm1 staining (red) was used to visualize differentiating ECs. White arrows indicate differentiating pre-ECs (*ISC*-GFP$^+$ and Pdm1$^+$cells). *ISC*-GFP$^+$ and Pdm1$^-$ cells are ISCs. *ISC*-GFP$^-$ and Pdm1$^+$ cells are mature ECs. (L) Quantification of the ratio of *ISC*-GFP$^+$ and Pdm1$^+$ cells per 10,000 μm$^2$ area of R4 region midguts from control *Drosophila* and *cad*-overexpressing *Drosophila* carrying *ISC$^{ts}$-Gal4*-driven *UAS-cad-HA* as shown in (J-K). The number n represents counted ROIs in midguts from each experiment. Each dot corresponds to one ROI (10,000 μm$^2$ area). (M-O) Immunofluorescence analyses of control (*FRT40A*, M), *cad$^2$* mutant (N), and *cad-HA* overexpressing (O) mosaic analysis with repressible cell marker (MARCM) clones (green) 7 days after clone induction (ACI) of flies. Prospero staining (red) was used to visualize EEs. White arrows indicate EEs in clones. DAPI stained nuclei are shown in blue. Scale bars represent 10 μm (S3A, S3B, S3D, S3E, S3G, S3H and S3J–S3K Fig), 5μm (S3M–S3O Fig). Error bars represent SD. Student's t-tests were used to assess statistical significance: *p < 0.05, **p < 0.01, ***p < 0.001, ****p < 0.0001, and NS (non-significant), which represents p > 0.05.
(TIF)

**S4 Fig. *Cad* regulates ISC-to-EC differentiation by modulating *Sox21a* expression.** (A-B) Western blotting of SOX21A-HA of whole midguts from young (7 days; A) and old (35 days; B) control *Drosophila* (*esg$^{ts}$-Gal4>UAS-GFP; Sox21a-HA*) and *Drosophila* carrying *esg$^{ts}$-Gal4>cad RNAi; Sox21a-HA*. Loading controls, *esg*-GFP (GFP) and a-tubulin (a-Tub). (C) Quantification of the relative band intensity of SOX21A-HA to *esg*-GFP as shown in experiments (A-B). (D) Western blotting of SOX21A-HA of whole midguts from young (7 days) and old (35 days) control *Drosophila* (*esg$^{ts}$-Gal4>UAS-GFP; Sox21a-HA*). Loading controls, *esg*-GFP (GFP) and a-tubulin (a-Tub). (E) Quantification of the relative band intensity of SOX21-A-HA to *esg*-GFP as shown in experiments (D). (F) Western blotting of SOX21A-HA of whole midguts carrying *Sox21a-HA* were treated with or without PQ (PQ-REC-1D). Loading controls, a-tubulin (a-Tub). (G) Relative mRNA fold changes of *Sox21a* in sorted *esg$^+$* cells from midguts of young (10 days) and old (50 days) *Drosophila* carrying *esg-GFP* and *Sox21a-HA*. The changes of expressions were plotted relative to the young *Drosophila*, which was set to 1. Error bars indicate the standard deviation (SD) of three independent experiments. (H) Immunofluorescence images of CAD-EGFP (green) and SOX21A-HA (red) staining with the midgut

section from PMG of young (5 days) and old (30 days) *Drosophila* carrying *cad-EGFP* and *Sox21a-HA*. The separated channels of GFP and HA were indicated in the lower panel. The enlarged insets show *Sox21a-HA*[+] cells (red) with CAD-GFP (green) staining. (I) Quantification of fluorescence intensity of CAD-EGFP and SOX21A-HA per HA[+] cell as shown in (H). The number n is indicated. Each dot represents one SOX21A-HA[+] cell. DAPI stained nuclei are shown in blue. Scale bars represent 10μm (S4H Fig). Error bars represent SD. Student's t-tests were used to assess significance: *p $< 0.05$, **p $< 0.01$, ***p $< 0.001$, ****p $< 0.0001$, and NS (non-significant), which represents p $> 0.05$.
(TIF)

**S5 Fig.** *GATAe* **functions as a downstream of cad to regulate intestinal stem cell to enterocyte differentiation.** (A-D) Immunofluorescence images of *esg*-GFP (green) and Pdm1 (red) staining with the midgut in homeostasis. The midgut section from PMG of control *Drosophila* (A, *esg*[ts]-Gal4>UAS-GFP), *Drosophila* carrying *esg*[ts]-Gal4>UAS-cad-HA (B), *Drosophila* carrying *esg*[ts]-Gal4>UAS-GATAe (C), and *Drosophila* carrying *esg*[ts]-Gal4>UAS-cad-HA and *UAS--GATAe* (D). *esg*-GFP (green) represents ISCs and their differentiating cells. Pdm1 staining (red) was used to visualize differentiating ECs. White arrows indicate differentiating pre-ECs (*esg*-GFP[+] and Pdm1[+] cells). *esg*-GFP[+] and Pdm1[-] cells are ISCs or EBs. *esg*-GFP[-] and Pdm1[+] cells are mature ECs. (E) Quantification of the ratio of *esg*-GFP[+] and Pdm1[+] cells per 10,000 μm[2] area of R4 region midguts of control *Drosophila* with genotypes as indicated in A-D. The number n represents counted ROI in midguts from each experiment. Each dot corresponds to one ROI (10,000 μm[2] area). DAPI stained nuclei are shown in blue. Scale bars represent 10μm (S5A–S5D Fig). Error bars represent SD. Student's t-tests were used to assess significance: *p $< 0.05$, **p $< 0.01$, ***p $< 0.001$, ****p $< 0.0001$, and NS (non-significant), which represents p $> 0.05$.
(TIF)

**S6 Fig. Activating** *hop* **functions could rescue the defect of ISC-to-EC differentiation caused by** *cad* **overexpression.** (A-D) Immunofluorescence images of *NRE*-GFP (green) and Pdm1 (red) staining with the midgut fed with PQ (PQ-REC-1D). The midgut section from the R4 region of control flies (A, *NRE*[ts]-Gal4-driven *UAS-GFP*), *Drosophila* carrying *NRE*[ts]-Gal4-driven *UAS-cad-HA* (B), *Drosophila* carrying *NRE*[ts]-Gal4-driven *UAS-hop*[TUM] (C), and *Drosophila* carrying *NRE*[ts]-Gal4-driven *UAS-cad-HA* and *UAS-hop*[TUM] (D). Pdm1 staining (red) was used to visualize differentiating ECs. White arrows indicate differentiating pre-ECs (*NRE*-GFP[+] and Pdm1[+] cells). *NRE*-GFP[+] and Pdm1[-] cells are EBs. *NRE*-GFP[-] and Pdm1[+] cells are matured ECs. (E) Quantification of the ratio of *NRE*-GFP[+] and Pdm1[+] cells Per 10,000 μm[2] area of the R4 region midguts as shown in (A-D). The number n represents counted regions of interest in midguts from each experiment. Each dot corresponds to one region of interest (ROI = 10,000 μm[2] area). DAPI stained nuclei are shown in blue. Scale bars represent 10μm (S6A–S6D Fig). Error bars represent SD. Student's t-tests were used to assess significance: *p $< 0.05$, **p $< 0.01$, ***p $< 0.001$, ****p $< 0.0001$, and NS (non-significant), which represents p $> 0.05$.
(TIF)

**S7 Fig. Reduction of** *cad* **expression in ISCs and progenitor cells represses age-associated gut hyperplasia in** *Drosophila.* (A-B) Quantification of the *esg*[+] cell number in AMG (B) or PMG (A) from 35-day-old (25 days at 18°C, then 10 days at 29°C) control *Drosophila* (*esg*[ts]-Gal4>UAS-GFP) and *Drosophila* carrying *esg*[ts]-Gal4>cad RNAi. The number n is indicated. Each dot corresponds to one ROI (10,000 μm[2] area). (C-D) Quantification of pH3[+] (C) or *esg*[+] (D) cell numbers in midguts from 35-day-old (25 days at 18°C, then 10 days at 29°C) control

*Drosophila* (*esg^ts^-Gal4>UAS-GFP*) and *Drosophila* carrying *esg^ts^-Gal4>UAS-cad-HA*. The number n represents the whole midguts in C. One dot corresponds to one midgut in C. The number n represents the ROI in midguts from each experiment in D. One dot corresponds to one ROI (10,000 μm$^2$ area) in D. (E-F) Immunofluorescence images of *esg*-GFP (green) and TUNEL (red) staining with the midgut section from PMG of 35-day-old (25 days at 18°C, then 10 days at 29°C) control *Drosophila* (E, *esg^ts^-Gal4>UAS-GFP*) and *Drosophila* carrying *esg^ts^-Gal4>cad RNAi* (F). *esg*-GFP (green) identifies ISCs and their differentiating cells. TUNEL staining (red) was used to visualize the apoptotic cells. White arrowheads indicate apoptotic cells. (G) Survival rate of control *Drosophila* (*esg^ts^-Gal4*-driven *UAS-GFP*) and *Drosophila* carrying *esg^ts^-Gal4*-driven *cad* RNAi. Adult flies were cultured at 18°C for 25 days and shifted to grow at 29°C to turn on the UAS-Gal4 system. The survival rates of these flies were recorded on the 25$^{th}$ day. DAPI stained nuclei are shown in blue. Scale bars represent 10 μm (S7A, S7B, S7K and S7L Fig). Error bars represent SD. Student's t-tests were used to assess significance. $^*$p < 0.05, $^{**}$p < 0.01, $^{***}$p < 0.001, $^{****}$p < 0.0001, and NS (non-significant), which represents p > 0.05.
(TIF)

**S1 Table. List of gene counts from RNA-seq, normalized by the DESeq2 R package.** Gene symbols and gene locations are indicated.
(XLSX)

**S2 Table. List of changed genes in a pair-wise comparison of *esg^ts^>UAS-lacZ Drosophila* to *esg^ts^>cad* RNAi *Drosophila*.** Gene symbols, gene locations, fold changes (i.e., logFC, log2 of the fold change), and P values (obtained by hypergeometric test) are indicated.
(XLSX)

**S3 Table. *Drosophila* lines used in this study.**
(XLSX)

## Acknowledgments

We thank the BDSC, the VDRC, and the THU for fly strains, and the DSHB for antibodies.

## Author Contributions

**Conceptualization:** Kun Wu, Haiyang Chen.

**Data curation:** Kun Wu, Yiming Tang, Qiaoqiao Zhang, Zhangpeng Zhuo, Jingping Huang, Jie'er Ye, Xiaorong Li.

**Formal analysis:** Kun Wu, Yiming Tang.

**Funding acquisition:** Haiyang Chen.

**Investigation:** Kun Wu, Yiming Tang, Qiaoqiao Zhang, Zhangpeng Zhuo.

**Methodology:** Kun Wu, Yiming Tang, Qiaoqiao Zhang, Zhangpeng Zhuo, Haiyang Chen.

**Project administration:** Kun Wu, Yiming Tang, Haiyang Chen.

**Resources:** Haiyang Chen.

**Software:** Kun Wu, Yiming Tang, Zhangpeng Zhuo.

**Supervision:** Haiyang Chen.

**Validation:** Kun Wu, Yiming Tang, Qiaoqiao Zhang, Zhangpeng Zhuo, Jie'er Ye.

**Visualization:** Kun Wu, Yiming Tang, Zhangpeng Zhuo.

**Writing – original draft:** Kun Wu, Yiming Tang, Zhiming Liu, Haiyang Chen.

**Writing – review & editing:** Kun Wu, Yiming Tang, Xiao Sheng, Zhiming Liu, Haiyang Chen.

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
