## [Decision Letter · Decision Letter 0]

25 Feb 2021

Dear Dr Chen,

Thank you very much for submitting your Research Article entitled 'Aging-related upregulation of the homeobox gene caudal represses intestinal stem cell differentiation in Drosophila' to PLOS Genetics. Our apologies for the delay in getting reviewers' comments back to you.

The manuscript was fully evaluated at the editorial level and by three independent peer reviewers. The reviewers appreciated the attention to an important problem, and each of them commented on the importance of the topic covered by your study. However, all three reviewers raised substantial concerns about the current manuscript. As you will read, each of the reviewers provided extensive comments. Based on the reviews, we will not be able to accept this version of the manuscript. Although we would be willing to review a much-revised version, given how extensive the comments are and the number of new experiments the reviewers find to be essential for publication, it may make more sense to completely revise and submit a new manuscript elsewhere. If you do decide to submit revisions to PLOS Genetics, it will of course have to go out for review again. We cannot, of course, promise publication at that time.

Should you decide to revise the manuscript for further consideration here, your revisions should address the specific points made by each reviewer. As you will see from the reviewers comments, there are quite a few new experiments that are suggested. A revised submission should address these by providing additional data, or by clearly stating why the suggested experiments cannot be done. Most importantly, please address control experiments that are either missing or not appropriate done, as detailed in the reviewers' comments. Please also pay extra attention to making the methods section more complete, fixing errors in figure legends and adding more complete citations. We will also require a detailed list of your responses to the review comments and a description of the changes you have made in the manuscript.

If you decide to revise the manuscript for further consideration at PLOS Genetics, please aim to resubmit within the next 60 days, unless it will take extra time to address the concerns of the reviewers, in which case we would appreciate an expected resubmission date by email to plosgenetics@plos.org.

[LINK]

We are sorry that we cannot be more positive about your manuscript at this stage. Please do not hesitate to contact us if you have any concerns or questions.

Yours sincerely,

Giovanni Bosco, Ph.D.

Associate Editor

PLOS Genetics

Hua Tang

Section Editor: Natural Variation

PLOS Genetics

Reviewer's Responses to Questions

**Comments to the Authors:**

Reviewer #1: In this manuscript, Wu and colleagues addressed the molecular mechanisms that underlie age-related decline in stem cell differentiation efficiency in Drosophila. Using the available genetic tool kits, authors identified the function of caudal (cad), a homeobox transcription factor in the differentiation of midgut intestinal stem cells (ISCs) to enterocytes (EC). Increased cad expression in ISCs and enteroblasts (EB) of aged flies prevents differentiation of progenitor cells by inhibiting the JAK/STAT-SOX21a-GATAe signaling pathway. Cell-specific knockdown of cad in the midgut ISCs and EBs increases differentiation efficiency, thereby restrain age-related hyperplasia in Drosophila.

Of fundamental interest, this study is important and will be of interest to broader readers of PLOS Genetics. I have a few suggestions and comments below that may be potentially useful for the authors to consider:

Major comments:

1. By employing a conditional knockdown approach, the authors present a key find that the reduction of cad in ISCs and progenitor cells increases differentiation of ISC to EC (Figure 2).

a. In these experiments, the authors used esgts-Gal4 to drive two reporter lines (UAS-GFP and UAS cad-RNAi). The authors should either report the RNAi-mediated knockdown efficiency or cite the previous references for the UAS cad-RNAi.

b. Lucidly report the conditional knockdown protocol in the methods sections. In methods, ‘The tub-Gal80ts transgenes Drosophila were grown at 18 ℃ and transferred to 29 ℃ for 7 days, followed by dissection’ and in figure legend 2, ‘….the midgut section from the R4 region of 9-day-old (2 days at 18 ℃, then 7 days at 29 ℃) control Drosophila….’

2. Pdm1 staining used to visualize differentiating ECs is inconsistent with high background signals. Do the authors have any comments? Also, the authors should include in detail methods for image acquisition, processing, and fluorescent intensity quantification.

3. It is interesting that esgts-Gal4-driven UAS cad-RNAi in the progenitor cells reduces gut hyperplasia in aged flies.

a. It would be useful to report the expression of JAK/STAT signaling molecules in the midgut of aged esgts-Gal4 > UAS cad-RNAi flies.

b. Does depletion of downstream signaling molecules JAK/STAT-SOX21a-GATAe reverts the midgut phenotype of esgts-Gal4 > UAS cad-RNAi in aged flies?

4. Some comments on the following sections for authors to consider:

a. In the methods section, more details should be provided and alternatively, if the procedures are previously described, then the reference citations should be included.

b. Reference citations for the genetic tools used in the manuscript are missing.

c. Authors should make sure that the genotypes are clear, correct, and consistent throughout the manuscript.

d. Figure legends are too wordy and lengthy - consider reducing the repetitive statements.

5. The conclusion that the increased cad expression in the progenitor cells represses age-associated ISCs differentiation in flies is well established. Discuss what and how cad expressions regulated in young and aged flies would be useful. Also, how does the decline in differentiation efficiency of the midgut cells affect the interorgan communication in flies?

Minor comments:

1. Drosophila is misspelled as Drosospila in several places.

2. Methods section corresponding to figure 1B is missing.

3. Methods section should include fly culturing and maintenance for the aging experiments.

4. Page 9, ‘….midguts of wild-type control flies’ should be ‘…..midguts of experimental/genotype control flies’

5. Page 34, repetition of the words: ‘White arrows indicate differentiating pre-ECs (both esg-GFP+ and Pdm1+ (esg-GFP+ and Pdm1+) cells)’

6. Figure 4A legend: ‘esgts-Gal4-driven UAS-GFP’ – according to methods and figure, reporter line is UAS-LacZ.

7. Figure 4C: ‘SOX21A-HA; cad RNAi’ should be ‘cad RNAi; SOX21A-HA’

8. Figure 6C: ‘UAS-cad-HA; 10xSTAT-GFP’ should be ‘10xSTAT-GFP; UAS-cad-HA’

9. Figure 7 legend: esgts-Gal4-driven cad-RNAi was induced from 26 day after fly eclosion, not at.

Reviewer #2: In this study, Wu and colleagues investigated the role of caudal (cad) in the differentiation of intestinal stem cells in Drosophila midguts. The authors identified that cad, a homeobox transcription factor, is upregulated in aged flies. Knockdown of cad in intestinal stem/progenitor cells promotes their differentiation towards enterocytes, whereas overexpression of cad prevents differentiation and regeneration upon injury. The author further determined that cad regulates ISC differentiation through Jak/STAT-Sox21a-GATAe signaling cascades. Lastly, the authors provide evidence suggesting reduction of cad in intestinal stem/progenitor cells prevent gut hyperplasia in aged Drosophila. Overall, this study is well-designed and the evidence strongly support the important role of cad in the regulation of ISC/progenitor cell differentiation. However, given the title and the beginning of the current manuscript, I expected to see data regarding the role of cad in both young and aged flies. It turns out a major portion of this study is focused on the function of cad in homeostasis or regeneration. Below are some major issues I believe the authors should address.

1. The authors first examined the expression levels of CAD-GFP in young and old flies. It seems to me that CAD is expressed in all cell types, perhaps at a higher level in ECs than in ISCs (Fig.1C). Based on the quantification of CAD-GFP intensity, the levels of CAD increase in all cell types upon aging. I believe the authors should clarify the expression pattern of CAD.

2. Is there antibody available to detect endogenous CAD? If so, it would be nice to show endogenous CAD levels increase in aged animals. If not, the authors should at least examine the expression of CAD-GFP in young and aged flies by western blot, which could support their claim using a different method.

3. Does depletion of cad in ISCs have an effect on their proliferation? The phenotype looks like a block of differentiation, but it would be important to rule out some other possibilities. Therefore, I suggest the authors should examine the proliferation rate of ISCs by phospho-Histone H3 staining.

4. Many of the functional experiments were done using RNAi-mediated knockdown, except Fig3J-M. Have the authors compare the effect of cad knockdown versus knockout?

5. The experiments of examine CAD function in ISC/progenitor cells were all done in young flies. Have the authors perform similar experiments in aged animals?

6. The authors mentioned in the beginning that cad expression in ISCs/EBs can be induced by bacteria infection. Similarly, does PQ-induced injury affect CAD expression levels in flies midguts?

7. The authors showed that overexpression of cad in ISC/progenitor cells prevent their differentiation to ECs. Are those ISC/progenitor cells able to proliferate upon injury? Also, is there an effect on the overall fitness of flies overexpressing cad in ISC/progenitors?

8. The authors examined SOX21A-HA expression levels upon cad depletion by immunofluorescence staining. I would help their claim if the authors can show the increase of protein levels by western blot. Also, it would be interesting to compare the SOX21 levels in young versus old flies.

9. It is not clear to me why the authors decided to investigate the interaction between GATAe and cad. Is GATAe one of the differentially expressed genes in their RNA-seq results? Given GATAe is downstream of Sox21a, one would naturally think GATAe is downstream of cad.

10. The authors showed knockdown of cad in ISC/progenitor cells prevent gut hyperplasia in aged flies. Does that have an effect on the overall fitness of aged flies or their longevity?

Reviewer #3: The report by Wu and colleagues investigates the role of the transcription factor Caudal (CAD) in the control of the Drosophila midgut intestinal lineage. The authors first confirm previous finding indicating that cad is upregulated in the gut of aged flies. Then, using a protein trap they show that although CAD protein expression increases in every cell type of the gut upon aging there is a greater upregulation in ISCs and EBs. The authors further show that inactivation of cad in EBs leads to the accumulation of cells expressing the EC marker, Pdm1, in normal condition of homeostasis while overexpression of cad decreases EC production after gut injury. Using classical Drosophila genetic tools and RNASeq, the authors bring some evidences suggesting that caudal regulates EB differentiation in EC upstream of the well described genetic framework composed of JAK-STAT/Sox21a/GATAe. Lastly, the authors find that a depletion of cad in the progenitor cells (ISC - EBs) leads to a decrease in hyperplasia formation in aged guts possibly because of the role of cad in the decline of ISC capacity to differentiate over time.

While a role for cad in midgut innate immune response and in ISC proliferation in midgut was already reported, its function in EBs to regulate differentiation and ECs formation was unexplored. Overall, the paper is clear and the conclusions are supported by quantified experiments. In my opinion, the weakness of the study at this stage is that the authors fail to completely demonstrate that the role of cad is restricted to EB-EC differentiation. Further work should discriminate between a role in the control of stem cell proliferation as reported in Choi et al 2008 and a role in EC differentiation upon homeostatic condition and after injury. The proposed model itself is quite convincing in normal condition of homeostasis (although not free of alternative explanations) but it is rather preliminary to extend it to aged guts and hyperplasia formation. Indeed, the data in old guts are limited to a partial phenotypical characterization of cad depletion in progenitor cells.

Here are some issues the authors should address before publication can be recommended:

Major Points:

1. The authors report that in the RNA-seq from Duta et al, cad is highly expressed in ISC and EB in regions R4-R5 but omitted to mention that cad is also highly expressed in EC and even more in EE in this region under normal condition of homeostasis.

2. Regarding CAD-EGFP expression in the different cell types of the midgut in Fig1: the authors limit their analysis to the region R4 in Fig1 C-N. It would be informative to report CAD-EGFP expression in the other regions of young and old guts. Is there any obvious difference as suggested by the RNA-seq data or is CAD uniformly distributed along the gut?

The authors propose that CAD modulates sox21a expression. sox21a expression has been reported in Zhai et al 2017 to be higher in ISC than in EB and to be totally shut down before the Notch activity reporter in the EBs. Here, looking at Fig 1E and 1H CAD-GFP seems to be less upregulated in ISCs (Dl+ cells) than in ISC/EB (esg+ cells) in general. It would be interesting to confirm that indeed CAD is differentially expressed between ISC and EB by co-staining of esg-lacZ/Dl/CADGFP or Dl/NRElacZ/CADGFP. Alternatively, it would be pertinent to compare Sox21a-HA and CAD-GFP expression profile in ISC/EBs of young and old guts. Is there a transient upregulation of CAD at the same time as Sox21 disappearance?

3. One of the main conclusions of the authors is the role of cad in the regulation of EB differentiation. I feel that the data backing up this conclusion are rather weak. Indeed, it is currently mainly based on a partial characterization of cad knockdown or overexpression in progenitor cells. This phenotypical characterization needs to be completed to rule out other hypothesis than a role of cad in EB differentiation.

It is essential for each condition to look at ISC proliferation using PH3 staining. This is important to rule out the possibility of an increase of ISC division leading to an excess of pre-EC production, expressing Pdm1 while maintaining some GFP due to protein perdurance. A decrease of ISC proliferation or ISC numbers could also explain a reduction of EC production upon overexpression of cad instead of a differentiation defect. This is also important since caudal has been proposed to regulate cell proliferation in the adult posterior midgut by Choi et al 2008.

Quantifications of the number of ISC (Dl+ cells) and esg+ cells are reported only for esgts>cad RNAi, and is limited to a ratio of Pdm1+ GFP+ / total GFP+ cells for the other conditions. In order to be able to properly interpret the results it would be important to have for each conditions de number of PH3+ cells, the number of Dl+ cells, and total GFP+ cells per region or per guts.

CAD appears to be enriched in regions R4 and R5 but is expressed in all the regions of the guts. The phenotypes reported here only mention R4 region. What about the other regions? Sox 21a appears to have different functions on ISC proliferation and progenitor differentiation in the anterior midgut or in the posterior midgut, is it true also for CAD?

4. The authors proposed that cad knockdown phenotypes are reminiscent of sox21a overexpression. In Zhai et al., 2015 sox21a overexpression leads to EB differentiation into EC but this is associated with a decrease in progenitor cell number (esg+ cells). This is different here, esg+ cell number remains the same upon cad RNAi expression. This is the number of ISCs (Dl+ cells) that decrease. The authors do not comment on this major difference. This should be discussed. This is not clear how after 7 days of cad knockdown a decrease of ISCs is observed but the number of GFP+ cells remains constant. Since the ISCs are the only dividing cell type in the midgut, how the same number of GFP+ cells can be produced while stem cells are lost? Is there an increase in the proliferation of the remaining ISC in order to compensate?

The authors do not address general cell death in these conditions. Is the ISC loss (Dl+ cells) due to differentiation as proposed by the authors or is it related to cell death? This could be tested both searching for increase ISC cell death in mutant tissue, and test a potential rescue in conditions preventing cell death.

5. The authors limit their clonal analysis of cad RNAi and cad overexpression phenotypes to a ratio of Pdm1+ cells. To properly interpret the data, it would be important to know at least how many cells per clone are found in each condition (indication of proliferation). Alone the ratio of Pdm1 + cell per clone is not very meaningful, a change in the number of Pdm1+ cells could be due to an increase or a decrease of ISC proliferation and not to a change in differentiation capacity. In Fig3L, clones seem much smaller upon overexpression of cad compared to control. Is the decrease of Pdm1+ cells due to less proliferation rather than a defect of differentiation? It would also be essential to have a better idea of the composition of the clones. Dl+ cells (number of ISC per clone), NRE lacZ to have an idea of EB accumulation and prospero+ cells to have the number of EE cell per clone. This is especially important since the depletion of sox21a (the proposed main downstream effector of cad) affects not only EC differentiation but also ISC proliferation and EE prospero + cells production.

6. Zhai et al., 2017 reported that esgts> sox21a guts form nest containing a slightly increase in the number of cells suggesting an additional role of sox21a in promoting ISC proliferation. Is it also something observed here for cad knockdown? Is it true also when cad RNAi is expressed specifically in the EB?

7. Regarding the overexpression of cad, why the authors only report its effect after PQ treatment. What is happening in homeostatic conditions? Proposed downstream effectors Sox21a, GATAe, JAK/Stat, all have been tested in normal condition of homeostasis in previous papers Zhai et al., 2015, 2017, Chen et al 2016….

In Fig.4G sox21a appears to be upregulated after damage, this is a bit contradictory with the data presented in Fig 1 and Sup1 indicating an upregulation of cad in aged gut and after Pe infection and with data from Chen et al 2016 where a downregulation of Sox21a is observed after DSS damage induction in the gut. The authors should test the impact of PQ and DSS treatments on cad expression level.

8. Since the authors proposed that sox21a is the main target of CAD in the progenitor cells of the midgut it would be interesting to compare their bulk RNAseq data with published RNAseq data for Sox21a overexpression by Zhai et al., 2017. Is there also an upregulation of expected downstream effectors such as pdm1, GATAe, Connectin, armadillo, E-cadherin…

9. Since cad appears to be required in the EB to regulate EB-EC differentiation, I am not sure I understand why further genetic interaction experiments and rescue experiments have been done using NREts driver instead of esgts. This specifically true for Sox21a-HA expression upon cad RNAi or UAS-sox21a rescue of UAS-cad-HA experiments in Fig4.

10. In rescue and genetic interaction experiments, some controls are missing making impossible to properly interpret the data. For example, in the case of the interaction between cad and Notch pathway, in Fig5 I-L esgts>N RNAi alone is missing. This is important because the double knockdown of N and cad appears to neither give a cad loss phenotype (increase of pdm1+ cells) nor the well described Notch loss phenotype (massive accumulation of ISC/EE and no EC). The conclusion appears more complicated than the one proposed by the authors. Does Notch act downstream, upstream or in parallel of cad? Similarly, in Fig4 H-K, esgts>UAS-sox21a phenotype alone is missing, in Fig5 A-D esgts>GATAe RNAi condition is missing, in Fig5 E-H NREts>GATAe RNAi condition is missing, in Fig6 D-G esgts>UAS-HopTUM alone is missing, in Fig6 H-K NREts>UAS-HopTUM alone is missing, in Fig6 L-O esgts>stat92E RNAi alone is missing.

11. The aging experiments are again difficult to interpret at this stage. The authors hypothesis that cad upregulation in aged gut contributes to a decrease of differentiation efficiency. Based on the data, it appears that cad knockdown prevent accumulation of extra esg+ cells as well as Dl+ cells in old guts. The authors do not address general cell death and proliferation in the tissue upon cad knockdown upon aging. Some quantifications should be added to determine whether there is a major effect on ISC survival and division in these conditions. It would be also interesting to evaluate the ratio of esg+ cells to total cells in the different conditions or maybe to induce late MARCM clones to test that indeed there is a change in EC production upon aging and after cad knockdown. Since cad appears to be required only in EB to regulate EC differentiation in young guts, the experiment should be repeated on old guts using NREts as a driver to confirm the cell type specific function of cad upon aging.

Since the upregulation of cad appears limited to R4-R5 regions, is this absence of hyperplasia limited to these regions or is it true all along the midgut? A regional quantification would be helpful. Finally, testing the impact of cad overexpression on aged guts seems to be an obvious experiment.

12. To extend the proposed model of caudal regulating EB differentiation in EC upstream of the framework composed of JAK-STAT/Sox21a/GATAe to old guts, it would be important to test if some changes of sox21a or GATAe expression are observed upon aging and if cad KD impacts them in old guts.

Minor points:

1. In Fig1 C, D , F, G, I, J, L, M it would be helpful to show split channel with GFP only not restricted to insets but for larger views of the region. Representative image of old and young whole midguts or whole post midguts would also help to properly follow CAD-GFP expression.

2. There are multiple errors in Fig. 1 legend:

. “(E) Quantification of fluorescence intensity of CAD-GFP in esg-LacZ+ cells of 7-day and 30-day-old Drosophila as shown in (B-C)” instead of (C-D).

. “(H) Quantification of fluorescence intensity of CAD-GFP in Dl+ ISCs of 7-day and 30- day-old Drosophila as shown in (E-F).” instead of (F-G).

. “(K) Quantification of fluorescence intensity of CAD-GFP in Pdm1+ ECs of 7-day and 30-day-old Drosophila as shown in (H-I)” instead of (I-J).

“(N) Quantification of fluorescence intensity of CAD-GFP in Pros+ EEs of 7-day and 30-day-old Drosophila as shown in (K-L)” instead of (M-M).

3. Details on RT-qPCR are lacking in material and methods:

Age of the guts, genotypes, cell sorting protocol and RT-qPCR.

4. Ref 22 correspond to mice crypt culture and not aged Drosophila midgut as mentioned in the text.

**Have all data underlying the figures and results presented in the manuscript been provided?**

Reviewer #1: Yes

Reviewer #2: Yes

Reviewer #3: Yes

PLOS authors have the option to publish the peer review history of their article (what does this mean?). If published, this will include your full peer review and any attached files.

Reviewer #1: No

Reviewer #2: No

Reviewer #3: No

---

## [Decision Letter · Decision Letter 1]

8 Jun 2021

Dear Dr Chen,

We are pleased to inform you that your manuscript entitled "Aging-related upregulation of the homeobox gene caudal represses intestinal stem cell differentiation in Drosophila" has been editorially accepted for publication in PLOS Genetics. Congratulations!

Yours sincerely,

Giovanni Bosco, Ph.D.

Associate Editor

PLOS Genetics

Hua Tang

Section Editor: Natural Variation

PLOS Genetics

Comments from the reviewers (if applicable):

Reviewer's Responses to Questions

**Comments to the Authors:**

Reviewer #1: In the revised manuscript the authors have satisfactorily addressed my comments and I recommend the publication of this manuscript in PLOS Genetics.

Reviewer #2: The authors have addressed my previous concerns in the revised manuscript. I believe the conclusions are significantly strengthened by these changes. Particularly, I appreciate the additional experiments carried out in aged flies, which strongly suggest the important role of cad during aging. I am convinced the revised manuscript will be a great interest to the general reader of Plos Genetics.

Reviewer #3: The revised manuscript is significantly improved. The authors have made conscientious effort in addressing the vast majority of my original comments/suggestions.

The new data for the most part support their previous conclusions more strongly. They've done more experiments to complement the phenotypical characterization of cad loss and overexpression and also to exclude a role in the control of progenitor proliferation or cell death. Moreover, they notably improved the characterization of cad role in aging. Overall, I am now supportive of publication.

I have one minor comment: in the abstract the following sentences have been mixed up in my opinion.

“This study investigated Drosophila midguts and identified an obvious upregulation of caudal (cad), which encodes a homeobox transcription factor. This factor is traditionally known as a central regulator of embryonic anterior-posterior body axis patterning in intestinal stem/progenitor cells upon aging.”

I might be wrong but maybe the authors meant:

“This study investigated Drosophila midguts and identified an obvious upregulation of caudal (cad) in intestinal stem/progenitor cells upon aging, which encodes a homeobox transcription factor. This factor is traditionally known as a central regulator of embryonic anterior-posterior body axis patterning.

**Have all data underlying the figures and results presented in the manuscript been provided?**

Reviewer #1: **No: **

Reviewer #2: Yes

Reviewer #3: None

PLOS authors have the option to publish the peer review history of their article (what does this mean?). If published, this will include your full peer review and any attached files.

Reviewer #1: No

Reviewer #2: No

Reviewer #3: No

**Data Deposition**

http://datadryad.org/submit?journalID=pgenetics&manu=PGENETICS-D-21-00033R1

**Press Queries**

---

## [Editor Report · Acceptance letter]

30 Jun 2021

PGENETICS-D-21-00033R1 

Aging-related upregulation of the homeobox gene caudal represses intestinal stem cell differentiation in Drosophila 

Dear Dr Chen, 

We are pleased to inform you that your manuscript entitled "Aging-related upregulation of the homeobox gene caudal represses intestinal stem cell differentiation in Drosophila" has been formally accepted for publication in PLOS Genetics! Your manuscript is now with our production department and you will be notified of the publication date in due course.

With kind regards,

Katalin Szabo

PLOS Genetics

On behalf of:
